**An improved regional coupled modeling system for Arctic sea ice simulation and prediction: a case study for 2018**

**Chao-Yuan Yang[1], Jiping Liu[2], Dake Chen[1]**

[1]School of Atmospheric Sciences, Sun Yat-sen University, and Southern Marine Science and Engineering Guangdong Laboratory (Zhuhai), Zhuhai, Guangdong, China

[2]Department of Atmospheric and Environmental Sciences, University at Albany, State University of New York, Albany, NY, USA

Corresponding author:

Chao-Yuan Yang (yangchy36@mail.sysu.eu.cn) and Jiping Liu (jliu26@albany.edu)

**Abstract**

The improved/updated Coupled Arctic Prediction System (CAPS) is evaluated by a set of
Pan-Arctic prediction experiments for the year 2018, which is built on Weather Research and
Forecasting model (WRF), the Regional Ocean Modeling System (ROMS), the Community
Ice CodE (CICE), and a data assimilation based on the Local Error Subspace Transform
Kalman Filter. We analyze physical process linking improved/changed physical
parameterizations in WRF, ROMS, and CICE to changes in the simulated Arctic sea ice state.
Our results show that the improved convection and boundary layer schemes in WRF result in
improved simulation in downward radiative fluxes and near surface air temperature, which
influences the predicted ice thickness. The changed tracer advection and vertical mixing
schemes in ROMS reduces the bias in sea surface temperature and changes ocean temperature
and salinity structure in the surface layer, leading to improved evolution of the predicted ice
extent (particularly correcting the late ice recovery issue in the previous CAPS). The improved
sea ice thermodynamics in CICE have noticeable influences on the predicted ice thickness. The
updated CAPS can better predict the evolution of Arctic sea ice during the melting season
compared with its predecessor, though the prediction still have some biases at the regional scale.
We further show that the updated CAPS can remain skillful beyond the melting season, which
may have potential values for stakeholders making decisions for socioeconomical activities in
the Arctic.

## 1. Introduction

Over the past few decades, the extent of Arctic sea ice has decreased rapidly and entered a thinner/younger regime associated with global climate change (e.g., Kwok, 2018; Serreze and Meier, 2019). The dramatic changes in the properties of Arctic sea ice have gained increasing attentions by a wide range of stakeholders, such as trans-Arctic shipping, natural resource exploration, and activities of coastal communities relying on sea ice (e.g., Newton et al., 2016). This leads to increasing demands on skillful Arctic sea ice prediction, particularly at seasonal timescale (e.g., Jung et al., 2016; Liu et al., 2019; Stroeve et al., 2014). However, Arctic sea ice prediction based on different approaches (e.g., statistical method and dynamical model) submitted to the Sea Ice Outlook, a community effort managed by the Sea Ice Prediction Network (SPIN, https://www.arcus.org/sipn), shows substantial biases in the predicted seasonal minimum of Arctic sea ice extent compared to the observations for most years since 2008 (Liu et al., 2019; Stroeve et al., 2014).

Recently, we have developed an atmosphere-ocean-sea ice regional coupled modeling system, for seasonal Arctic sea ice and climate prediction (Yang et al., 2020, hereafter Y20), in which the Los Alamos Sea Ice Model (CICE) is coupled with the Weather Research and Forecasting Model (WRF) and the Regional Ocean Modeling System (ROMS), hereafter called Coupled Arctic Prediction System (CAPS). To improve the accuracy of initial sea ice conditions, CAPS employs an ensemble-based data assimilation system to assimilate satellite-based sea ice observations. Seasonal Pan-Arctic sea ice predictions with improved initial sea ice conditions conducted in Y20 have shown that CAPS has the potential to provide skillful

Arctic sea ice prediction at seasonal timescale.
We know that the changes of sea ice variables (e.g., ice extent, ice concentration, ice
thickness, ice drift) are mainly driven by forcings from the atmosphere and the ocean.
Atmospheric cloudiness and related radiation influence surface ice melting (Huang et al., 2019;
Kapsch et al., 2016; Kay et al., 2008) and the energy stored in the surface mixed layer that
determines the seasonal ice melt and growth (e.g., Perovich et al., 2011, 2014). Atmospheric
circulation is the primary driver for the transportation of sea ice and partly responsible for the
variability of Arctic sea ice (e.g., Mallett et al., 2021; Ogi et al., 2010; Zhang et al., 2008).
Olonscheck et al. (2019) suggested that atmospheric temperature fluctuations explain a
majority of Arctic sea ice variability while other drivers (e.g., surface winds, and poleward heat
transport) account for about 25% of Arctic sea ice variability. The oceanic heat inputs (as well
as salt inputs) into the Arctic Ocean include the Atlantic Water (AW; Aagaard, 1989;
McLaughlin et al., 2009) through the Barents Sea, and the Pacific Water (PW; Itoh et al., 2013;
Woodgate et al., 2005) from the Bering Strait. The oceanic heat inputs from AW and PW are
not directly available for sea ice since they are separated from a cold and fresh layer underlying
sea ice (e.g., Carmack et al., 2015, Fig. 2). Vertical mixing by the internal wave (e.g., Fer, 2014)
and double diffusion (e.g., Padman and Dillon, 1987; Turner, 1973) are the principal processes
for upward heat transport from the subsurface layer (i.e., AW and PW) to the surface mixed
layer in the Arctic Ocean. Sea ice thermodynamics determines how thermal properties of sea
ice (e.g., temperature, salinity) change. These changes then influence the thermal structure of
underlying ocean through interfacial fluxes (i.e., heat, salt and freshwater fluxes; DuVivier et
al., 2021; Kirkman IV and Bitz, 2011) and ice thickness (e.g., Bailey et al., 2020).

The CAPS is configured for the Arctic with sufficient flexibility. That means each model

component of CAPS (WRF, ROMS, and CICE) has different physics options for us to choose
and capability to integrate ongoing improvements in physical parameterizations. Recently, the
WRF model has adapted improved convection and boundary layer schemes in the Rapid
Refresh (RAP) model operational at the National Centers for Environmental Prediction (NCEP,
Benjamin et al., 2016). The first question we want to answer in this paper is to what extent
these modifications can improve atmospheric simulations in the Arctic (i.e., radiation,
temperature, humidity, and wind), and then benefit seasonal Arctic sea ice simulation and
prediction. The ROMS model provides several options for tracer advection schemes. These
advection schemes can have different degrees of oscillatory behavior (e.g., Shchepetkin and
McWilliams, 1998). The oscillatory behavior can have impacts on sea ice simulation through
ice-ocean interactions (e.g., Naughten et al., 2017). The second question we want to answer in
this paper is to what extent different advection schemes can change the simulation of upper
ocean thermal structure and then Arctic sea ice prediction. Several recent efforts have
incorporated prognostic salinity into sea ice models. The CICE model has a new mushy-layer
thermodynamics parameterization that includes prognostic salinity and treats sea ice as a two-
phase mushy layer (Turner et al., 2013). Bailey et al. (2020) showed that the mushy-layer
physics has noticeable impacts on Arctic sea ice simulation within the Community Earth
System Model version 2. The third question we want to answer in this paper is whether the
mushy-layer scheme can produce noticeable influence on seasonal Arctic sea ice prediction.
Currently, SIPN focuses on Arctic sea ice predictions during the melting season, particularly
seasonal minimum. It is not clear that how predictive skills of dynamical models participating
in SIPN may change for longer period, i.e., extending into the freezing up period, which also
have significance on socioeconomic aspects. The assessment of the skills of global climate
models (GCMs) in predicting Pan-Arctic sea ice extent with suites of hindcasts suggested that
GCMs may have skill at lead times of 1-6 months (e.g., Blanchard-Wrigglesworth et al., 2015;
Chevallier et al., 2013; Guemas et al., 2016; Merryfield et al., 2013; Msadek et al., 2014;
Peterson et al., 2015; Sigmond et al., 2013; Wang et al., 2013; Zampieri et al., 2018). Moreover,
some studies using a "perfect model" approach, which treats one member of an ensemble as
the truth (i.e., assuming the model is prefect without bias) and analyzes the skill of other
members in predicting the response of the "truth" member (e.g., Meehl et al., 2007), suggested
that Arctic sea ice cover can be potentially predictable up to two years in advance (e.g.,
Blanchard-Wrigglesworth et al., 2011; Blanchard-Wrigglesworth and Bushuk, 2018; Day et al.,
2016; Germe et al., 2014; Tietsche et al., 2014). The last question we want to answer in this
paper is whether CAPS has predictive skill for longer periods (up to 7 months).

This paper is structured as follows. Section 2 provides a brief overview of CAPS,

including model configurations and data assimilation procedures. Section 3 describes the
designs of the prediction experiments for the year of 2018 based on major improvements/
changes in the model components compared to its predecessor described in Y20, examines the
performance of the updated CAPS, and offers physical links between Arctic sea ice changes
and improved/changed physical parameterizations. Section 4 discusses the predictive skill of
CAPS at longer timescale. Discussions and concluding remarks are given in section 5.
**2.    Coupled Arctic Prediction System (CAPS)**
As described in Y20, the CAPS has been developed by coupling the Community Ice CodE
(CICE) with the Weather Research and Forecasting Model (WRF) and the Regional Ocean
Modeling System (ROMS) based on the framework of the Coupled Ocean-Atmosphere-Wave-
Sediment Transport (Warner et al., 2010). The general description of each model component in
CAPS is referred to Y20. The advantage of CAPS is its model components have a variety of
physics for us to choose and capability to integrate follow-up improvements of physical
parameterizations. With recent achievements of community efforts, we update CAPS based on
newly-released WRF, ROMS, and CICE models. During this update, we focus on the Rapid
Refresh (RAP) physics in the WRF model, the oceanic tracer advection scheme in the ROMS
model, sea ice thermodynamics in the CICE model (see details in section 3), and investigate
physical process linking them to Arctic sea ice simulation and prediction. The same physical
parameterizations described in Y20 are used here for the control simulation (see Table 1). Major
changes in physical parameterizations as well as the model infrastructure in the WRF, ROMS,
and CICE models are described in section 3.
As described in Y20, the Parallel Data Assimilation Framework (PDAF, Nerger and Hiller,
2013) was implemented in CAPS, which provides a variety of optimized ensemble-based
Kalman filters. The Local Error Subspace Transform Kalman Filter (LESTKF; Nerger et al.,
2012) is used to assimilate satellite-observed sea ice parameters. The LESTKF projects the
ensemble onto the error subspace and then directly computes the ensemble transformation in
the error subspace. This results in better assimilation performance and higher computational
efficiency compared to the other filters as discussed in Nerger et al. (2012).
The initial ensembles are generated by applying the second-order exact sampling (Pham,
2001) to simulated sea ice state vectors (ice concentration and thickness) from an one-month
free run, and then assimilating sea ice observations, including: 1) the near real-time daily Arctic
sea ice concentration processed by the National Aeronautics and Space Administration (NASA)
Team algorithm (Maslanik and Stroeve, 1999) obtained from the NSIDC
(https://nsidc.org/data/NSIDC-0081/), and 2) a combined monthly sea ice thickness derived
from the CryoSat-2 (Laxon et al., 2013; obtained from http://data.seaiceportal.de), and daily
sea ice thickness derived from the Soil Moisture and Ocean Salinity (SMOS; Kaleschke et al.,
2012; Tian-Kunze et al., 2014; obtained from https://icdc.cen.uni-hamburg.de/en/l3c-smos-
sit.html). To address the issue that sea ice thickness derived from CyroSat-2 and SMOS are
unavailable during the melting season, the melting season ice thickness is estimated based on
the seasonal cycle of the Pan-Arctic Ice Ocean Modeling and Assimilation System (PIOMAS)
daily sea ice thickness (Zhang and Rothrock, 2003).
Different from Y20, in this study, we change the localization radius from 2 to 6 grids
during the assimilation procedures to reduce some instability during initial Arctic sea ice
simulations associated with 2 localization radii. As shown in Supplementary Figure S1, the ice
thickness with 2 localization radii and 1.5 m uncertainty (used in Y20) shows some
discontinuous features (Fig. S1a), which tends to result in numerical instability during the
initial integration. Such discontinuous features are obviously corrected with 6 localization radii
and 0.75 m uncertainty (Fig. S1b). Following Y20, here we test the 2018 prediction experiment
with 6 localization radii for the data assimilation, which shows very similar temporal evolution
of the total Arctic sea ice extent for the July experiment relative to that of Y20, although it (red
solid line) predicts slightly less ice extent than that of Y20 (blue line) (Supplementary Figure
S2). In this study, this configuration is designated as the reference for the following assessment
of the updated CAPS (hereafter Y20_MOD).
For the evaluation of Arctic sea ice prediction, Sea Ice Index (Fetterer et al., 2017;
obtained from https://nsidc.org/data/G02135) is used as the observed total sea ice extent, and
the NSIDC sea ice concentrations (SIC) derived from Special Sensor Microwave
Imager/Sounder (SSMIS) with the NASA Team algorithm (Cavalieri et al., 1996; obtained from
https://nsidc.org/data/nsidc-0051) is also used. For the assessment of the simulated atmospheric
and oceanic variables, the ECMWF reanalysis version 5 (ERA5; Hersbach et al., 2020;
obtained from https://cds.climate.copernicus.eu) and National Oceanic and Atmospheric
Administration (NOAA) Optimum Interpolation (OI) Sea Surface Temperature (SST)
(Reynolds      et      al.,      2007;      obtained      from
https://psl.noaa.gov/data/gridded/data.noaa.oisst.v2.highres.html)  are  utilized.  For  the
comparison of spatial distribution, SIC, ERA5, and OISST are interpolated to the model grid.
**3.  Evaluation of updated CAPS**
**3.1. Experiment designs and methodology**
The model domain includes 319 (449) x- (y-) grid points with a ~24 km grid spacing for
all model components (see Figure 2 in Y20). The WRF model uses 50 vertical levels, the
ROMS model uses 40 vertical levels, and the CICE model uses 7 ice layers, 1 snow layer, and
5 categories of sea ice thickness. The coupling frequency across all model components is 30
minutes. Initial and boundary conditions for the WRF and ROMS models are generated from
the Climate Forecast System version 2 (CFSv2, Saha et al., 2014) operational forecast archived
at NCEP (http://nomads.ncep.noaa.gov/pub/data/nccf/com/cfs/prod/). Sea ice initial conditions
are generated from the data assimilation described in section 2. Ensemble predictions with 8
members are conducted. A set of numerical experiments for the Pan-Arctic seasonal sea ice
prediction with different physics, starting from July 1$^{st}$ to October 1$^{st}$ for the year of 2018, has
been conducted. Table 2 provides the details of these experiments that allow us to examine
physical process linking improved/changed physical parameterizations in the updated CAPS
to Arctic sea ice simulation and prediction.
In this study, sea ice extent is calculated as the sum of area of all grid cells with ice
concentration greater than 15%. Besides the total Arctic sea ice extent, we also calculate the
ice extent for the following subregions: 1) Beaufort and Chukchi Seas (120°W-180, 60°N-
80°N), 2) East Siberian and Laptev Seas (90°E-180, 60°N-80°N), and 3) Barents, Kara, and
Greenland Seas (30°W-90°E, 60°N-80°N). To further assess the predictive skill of Arctic sea
ice predictions, we show the climatology prediction (CLIM, the period of 1998-2017) and the
damped anomaly persistence prediction (DAMP). Following Van den Dool (2006), the DAMP
is generated from the initial sea ice extent anomaly (relative to the 1998-2017 climatology)
scaled by the autocorrelation and the ratio of standard deviation between different lead times
and initial times (see the DAMP equation in Y20).
In order to understand physical contributors that drive the evolving Arctic sea ice state,
the mass budget of Arctic sea ice for all experiments is analyzed in this study as defined in
Notz et al. (2016, Append. E), including 1) sea ice growth in supercooled open water (frazil ice
formation), 2) basal growth, 3) snow-to-ice conversion, 4) top melt, 5) basal melt, 6) lateral
melt, and 7) dynamics process.
**3.2. Impacts of the RAP physics in the WRF model**
To examine the performance of the upgrades of physical parameterization in component
models in CAPS one step at a time compared to its predecessor in Y20, we define the
Y21_CTRL experiment that uses the RAP physics in the WRF model (see Table 2 for
differences between Y21_CTRL and Y20_MOD). Recently, the Rapid Refresh (RAP) model,
a high-frequency weather prediction/assimilation modeling system operational at the National
Centers for Environmental Prediction (NCEP), has made some improvements in the WRF
model physics (Benjamin et al., 2016), including improved Grell-Freitas convection scheme
(GF) and Mellor-Yamada-Nakanishi-Niino planetary boundary layer scheme (MYNN). For the
GF scheme, the major improvements relative to the original scheme (Grell and Freitas, 2014)
include: 1) a beta probability density function used as the normalized mass flux profile for
representing height-dependent entrainment/detrainment rates within statistical-averaged deep
convective plumes, which is given as:
$$Z_{u,d}(r_k) = cr_k^{\alpha} - (1 - r_k)^{\beta} - 1$$

where $Z_{u,d}$ is the mass flux profiles for updrafts and downdrafts, c is a normalization constant,

$r_k$ is the location of the mass flux maximum, $\alpha$ and $\beta$ determine the skewness of the beta

probability density function, and 2) the ECMWF approach used for momentum transport due

to convection (Biswas et al. 2020; Freitas et al. 2018; 2021). For the MYNN scheme, the RAP

model improves the mixing-length formulation, which is designed as:

$$\frac{1}{l_m} = \frac{1}{l_s} + \frac{1}{l_t} + \frac{1}{l_b}$$

where $l_m$ is the mixing length, $l_s$ is the surface length, $l_t$ is the turbulent length, and $l_b$ is
the buoyancy length. Compared to the original scheme, the RAP model changed coefficients
in the formulation of $l_s$, $l_t$, and $l_b$ for reducing the near-surface turbulent mixing, and the
diffusivity of the scheme. The RAP model also removes numerical deficiencies to better
represent subgrid-scale cloudiness (Benjamin et al. 2016, see Append. B) compared to the
original scheme (Nakanishi and Nino, 2009). In addition, some minor issues in the Noah land
surface model (Chen and Dudhia, 2001) have been fixed, including discontinuous behavior for
soil ice melting, negative moisture fluxes over glacial, and associated with snow melting.

Apparently, the above RAP physics can have influence on the behavior of simulated

atmospheric thermodynamics (i.e., radiation, temperature). Figure 1 and 2 show the spatial
distribution of the ERA5 surface downward solar and thermal radiation (SWDN and LWDN),
the prediction errors (ensemble mean minuses ERA5) of Y20_MOD, and the difference
between Y21_CTRL and Y20_MOD. For July, Y20_MOD (Fig. 1d) results in less SWDN over
most of ocean basins as well as Alaska and northeast US, western Siberia, and eastern Europe,
but more SWDN over southern and eastern Siberia compared with ERA5. For August and
September (Fig. 1e-f), the spatial distribution is generally similar to that of July, except that
eastern Siberia (less SWDN) and northern Canada (more SWDN) in August. It appears that the
magnitude of the prediction errors tends to decrease over the areas with large prediction errors
as the prediction time increases (i.e., July vs. September). Compared with Y20_MOD, the RAP
physics in Y21_CTRL result in large areas with smaller prediction errors in July (e.g., the
positive difference between Y21_CTRL and Y20_MOD reduces the negative prediction errors
in Y20_MOD), except the north Pacific (especially the Sea of Okhotsk) and north Canada (Fig.
1g). For August and September (Fig. 1h, i), encouragingly, there are more areas with smaller
prediction errors.
In contrast to SWDN, the prediction errors of LWDN in Y20_MOD has much smaller
magnitude (up to 100 $W/m^2$ in SWDN vs. 50 $W/m^2$ in LWDN) for the entire prediction period
(Fig. 2d-f). For July, Y20_MOD (Fig. 2d) simulates less LDWN over most of the model domain
compared with ERA5, except the Atlantic sector and north Greenland. For August, the areas
with negative prediction errors expand and the magnitude of prediction errors increases
(particularly in southeastern Siberia and northeast US) compared to that of July (Fig. 2e). For
September (Fig. 2f), the spatial distribution of LWDN is mostly similar to that of July, except
that north Canada and Canadian Archipelago show positive prediction errors. The Y21_CTRL
experiment with the RAP physics tends to reduce the prediction errors in Y20_MOD, especially
over eastern Siberia and the Atlantic sector in July to September (Fig. 2g-i).
Figure 3 shows the spatial distribution of the ERA5 2m air temperature, the prediction
errors of Y20_MOD, and the difference between Y21_CTRL and Y20_MOD. For Y20_MOD,
the predicted air temperature in July has small cold prediction errors over all ocean basins,
small-to-moderate cold prediction errors (~3-5 degrees) over Canada and Siberia, and
moderate-to-large cold prediction errors (~6-9 degrees) over eastern Europe (Fig. 3d). In
August (Fig. 3e), the cold prediction errors over most of the model domain are increased, in
particular, very large cold prediction error (over 10 degrees) is located over east Siberia. In
September, these cold prediction errors are decreased relatively, and some warm prediction
errors are found in north of Greenland (Fig. 3f). With the adaptation of the RAP physics in the
WRF model, Y21_CTRL, in general, produces a warmer state in most of the model domain
compared to that of Y20_MOD during the entire prediction period. For July (Fig. 3g), the
predicted air temperature is slightly warmer over the Arctic Ocean, the Pacific, and Atlantic
sectors, moderately warmer (~1-2 degrees) over central and eastern Siberia and Canadian
Archipelago, but the slightly colder over northern Canada than that of Y20_MOD. For August
and September (Fig. 3h), most of the model domain is warmer in Y21_CTRL than that of
Y20_MOD, in particular excessive cold prediction errors shown in Y20_MOD over Siberia are
reduced notably (~2.5-4 degrees). We notice that the RAP physics does not have significant
impacts on atmospheric circulation, given that Y21_CTRL and Y20_MOD have very similar
wind pattern (not shown).
Figure 4 shows the temporal evolution of the ensemble mean of the predicted Arctic sea
ice extent along with the NSIDC observations. In terms of the total ice extent, compared to the
Y20_MOD experiment (blue line), the Y21_CTRL experiment (yellow line) produces ~0.5
million km$^2$ more ice extent at the initial. Note that the difference in the initial ice extent is
related to that sea ice fields in Y20_MOD and Y21_CTRL (as well as other experiments listed
in Table 2) are initialized based on one-month free runs (section 2), which use different physical
configurations listed in Table 2. These one-month free runs do not have the same evolution in
sea ice fields and result in different initial ice fields after data assimilation. The ice extent in
Y21_CTRL decreases faster than Y20_MOD during the first 2-week integration. After that,
they track each other closely, and predict nearly the same minimum ice extent (~4.3 million
$km^2$). Like Y20_MOD, Y21_CTRL still has a delayed ice recovery in late September compared
to the observation. Compared with the CLIM/DAMP predictions (black dashed and dotted
lines), both Y20_MOD and Y21_CTRL have smaller prediction errors in August, but
comparable prediction errors after early September.

The difference in sea ice extent becomes larger at regional scales, in the East Siberian-

Laptev Seas, Y20_CTRL shows faster ice decline after mid-July than that of Y21_MOD,
whereas in the Beaufort-Chukchi Seas, Y21_CTRL predicts slower ice retreat after late July
than that of Y20_MOD (Fig. 4a, 4b). They are consistent with that Y21_CTRL predicts warmer
(relatively colder) temperature than that of Y20_MOD in the East Siberian-Laptev (Beaufort-
Chukchi) Seas. Both Y20_MOD and Y21_CTRL agree well with the observations in the
Barents-Kara-Greenland Seas (Fig. 4c). Compared with the observations, Y20_MOD performs
relatively better in regional ice extents than that of Y21_CTRL. Figure 5 shows the spatial
distribution of the NSIDC sea ice concentration and the difference between the predicted ice
concentration and the observations for all grid cells that the predictions and the observations
both have at least 15% ice concentration. The vertical and horizontal lining areas represent
difference of the ice edge location. Like the regional ice extent shown in Figure 4, Y21_CTRL
predicts lower (higher) ice concentration along the East Siberian-Laptev (Beaufort-Chukchi)
Seas (Fig. 5e$_1$-e$_3$). Y21_CTRL also predicts less ice in the central Arctic Ocean in August and
September, which is consistent with warmer temperature in Y21_CTRL relative to Y20_MOD.

Figure 6 shows the evolution of sea ice mass budget terms of Y20_MOD and Y21_CTRL,

averaged with cell-area weighting over the entire model domain. During the entire prediction
period, most of the ice loss in Y20_MOD and Y21_CTRL are caused by basal melting. The
surface melting has relatively small contribution in the total ice loss and mainly occurs in July.
However, compared with Y20_MOD (Fig. 6a), Y21_CTRL (Fig. 6b) shows much larger
magnitude for basal and surface melt. In a fully coupled predictive model, the changes of sea
ice are determined by the fluxes from the atmosphere above and the ocean below. Associated
with the increased downward radiation of the above RAP physics, Y21_CTRL absorbs more
shortwave radiation (SWABS, Fig. 7a) and allows more penetrating solar radiation into the
upper ocean below sea ice (SWTHRU, Fig. 7b) than that of Y20_MOD, especially in July. This
explains why Y21_CTRL has larger magnitude of surface and basal melting terms. Although
Y21_CTRL show larger magnitude in surface and basal melting than that of Y20_MOD, the
ice extent in Y21_CTRL and Y20_MOD shown in Figure 4 show similar evolution. The effect
of larger surface and basal melting in Y21_CTRL is largely reflected in the ice thickness change.
As shown in Figure S3, Y21_CTRL has thinner ice thickness than that of Y20_MOD, in the
East Siberian-Laptev Seas in July and in the much of central Arctic Ocean in August and
September.

**3.3. Impacts of the tracer advection in ROMS model**
Currently, the ROMS model that uses a generalized topography-following coordinate has
two vertical coordinate transformation options:

348
$$z(x,y,\sigma,t) = S(x,y,\sigma) + \zeta(x,y,t)\left[1 + \frac{S(x,y,\sigma)}{h(x,y)}\right] \quad (1)$$
$$S(x,y,\sigma) = h_c\sigma + [h(x,y) - h_c]C(\sigma)$$

or

$$z(x,y,\sigma,t) = \zeta(x,y,t) + [\zeta(x,y,t) + h(x,y)]S(x,y,\sigma)$$
$$S(x,y,\sigma) = \frac{h_c\sigma + h(x,y)C(\sigma)}{h_c + h(x,y)} \quad (2)$$

where $S(x,y,\sigma)$ is a nonlinear vertical transformation function, $\zeta(x,y,t)$ is the free-surface,
$h(x,y)$ is the unperturbed water column thickness, $C(\sigma)$ is the non-dimensional, monotonic,
vertical stretching function, and $h_c$ controls the behavior of the vertical stretching. In Y20, we
used the transformation 1 and the vertical stretching function introduced by Song and
Haidvogel (1994). However, the vertical transformation 1 has an inherent limitation for the
value of $h_c$ (expected to be the thermocline depth), which must be less than or equal to the
minimum value in $h(x,y)$. As a result, $h_c$ was chosen as 10 meters due to the limitation of
the minimum value in $h(x,y)$ in Y20. This limitation is removed with the vertical
transformation 2 and the vertical stretching function introduced by Shchepetkin (2010), and
$h_c$ can be any positive value. Here the Y21_VT experiment is conducted to examine the impact
of the vertical transformation in the ROMS model on seasonal Arctic sea ice simulation and
prediction, which uses the vertical transformation 2, the Shchepetkin stretching function, and
300 meters for $h_c$. As shown in Supplementary Figure S4-S5, compared to Y21_CTRL,
Y21_VT is less sensitive to the bathymetry and its layers are more evenly-distributed in the
upper 300 meters. With the changes of vertical layers of the upper ocean, the Y21_VT
experiment has minor SST changes relative to Y21_CTRL. The simulated temporal evolution
of total ice extent of Y21_VT (Fig. 4, red line) resembles to that of Y21_CTRL (Fig. 4, yellow
line), although some differences are seen at the regional scale in the areas with shallow water
(e.g., East Siberian, Laptev, Barents, and Kara Seas). The configuration of Y21_VT is used in
the following experiments.

It has been recognized that the tracer advection and the vertical mixing schemes have

important effects on ocean and sea ice simulation (e.g., Liang and Losch, 2018; Naughten et
al., 2017). Here the Y21_RP experiment is designated to explore the influence of different
advection schemes in the ROMS model. Specifically, the tracer advection scheme is changed
from the Multidimensional positive definite advection transport algorithm (MPDATA;
Smolarkiewicz, 2006) to the third-order upwind horizontal advection (U3H; Rasch, 1994;
Shchepetkin, and McWilliams, 2005) and the fourth-order centered vertical advection schemes
(C4V; Shchepetkin, and McWilliams, 1998; 2005). The MPDATA scheme applied in
Y20_MOD, Y21_CTRL, and Y21_VT is a non-oscillatory scheme but a sign preserving
scheme (Smolarkiewicz, 2006) that means MPDATA is not suitable for tracer fields having
both positive and negative values (i.e., temperature with degree Celsius in the ROMS model).
The upwind third-order (U3H) scheme used in Y21_RP is an oscillatory scheme but it
significantly reduces oscillations compared to other centered schemes (e.g., Hecht et al., 2000;
Naughten et al., 2017) available in the ROMS model.

Figure 8 shows the spatial distribution of the SST changes of Y21_VT and Y21_RP

relative to Y21_CTRL (as well as the OI SST and the difference between Y21_CTRL and
OISST). In general, Y21_CTRL shows cold prediction errors in the North Pacific (~2 degrees)
and the Atlantic (~3 degrees) compared to that of OISST in July, and these cold prediction
errors are enhanced as the prediction time increases (to 3-5 degrees, Fig. 8d-f). With the
U3H/C4V tracer advection scheme in Y21_RP, cold prediction errors shown in Y21_CTRL are
reduced significantly in the north Pacific and Atlantic, but SST under sea ice in much of the
Arctic Ocean is slightly colder than that of Y21_CTRL (Fig. 8j-l).
Y21_RP (Fig. 4, green line) shows comparable temporal evolution of the ice extent as
Y21_CTRL (as well as Y21_VT) until near the end of July. After that, the ice melting slows
down (closer to the observation) and the ice extent begins to recover earlier (after the first week
of September) in Y21_RP compared to Y21_CRTL. This leads to much smaller prediction error
in seasonal minimum ice extent relative to the observation. Y21_RP also shows better
predictive skill after late August compared with the CLIM/DAMP predictions (black dashed
and dotted lines). This suggests the delayed ice recovery in late September shown in Y20_MOD,
Y21_CTRL and Y21_VT is in part due to the choice of ocean advection and vertical mixing
schemes, which change the behavior of ocean state. At the regional scale, the slower ice decline
after July and earlier recovery of the ice extent in September mainly occur in the Beaufort-
Chukchi and Barents-Kara-Greenland Seas compared to that of Y21_CTRL (Fig. 4a, c). By
using U3H/C4V scheme, the Y21_RP experiment simulates higher sea ice concentration than
that of Y21_VT (Fig. 5$f_1$-$f_3$). For September, the Y21_RP experiment better predicts the ice
edge location in the Atlantic sector of the Arctic (i.e., smaller areas with horizontal/vertical
lining) compared to the experiments described above (not shown).

Figure 9 shows the evolution of sea ice mass budget terms of Y21_VT and Y21_RP.

Relative to Y21_VT, Y21_RP (with U3H/C4V scheme) results in increased frazil ice formation
in July, which is partly compensated by increased surface melting. Y21_RP also leads to
increased basal growth in mid- and late September (Fig. 9a, b).

Figure 10 shows the difference in the vertical profile of ocean temperature and salinity in

the upper 150 m averaged for the central Arctic Ocean between Y21_RP and Y21_VT. The
ocean temperature in the surface layer of Y21_RP is slightly colder during the prediction period
compared to that of Y21_VT (Fig. 10a), especially in August and September. Moreover, the
water in the surface layer (0-20 m) of Y21_RP is fresher than that of Y21_VT (Fig. 10b). They
reduce the freezing temperature and favor frazil ice formation. In the CAPS, the frazil ice
formation is determined by the freezing potential, which is the vertical integral of the difference
between temperature in upper ocean layer and the freezing temperature in the upper 5 m-layer.
The supercooled water is adjusted based on the freezing potential to form new ice and rejects
brine into the ocean that leads to saltier water between 20-50 m in Figure 10. It should be noted
that the increased frazil ice formation in July in Y21_RP might be also partly due to the
oscillatory behavior of U3H scheme, which makes the temperature fall below the freezing point
and then instantaneously forms new ice (as well as temperature/salinity adjustments).

**3.4. Impacts of sea ice thermodynamics in the CICE model**

In Y20, we used sea ice thermodynamics introduced by Bitz and Lipscomb (1999;

hereafter BL99) as the setup of CAPS, which assumes a fixed vertical salinity profile based on
observations. The new CICE model includes a MUSHY-layer ice thermodynamics introduced
by Turner et al. (2013), which simulates vertically and time-varying prognostic salinity and
associated effects on thermodynamic properties of sea ice. In the Y21_MUSHY experiment,
we change the ice thermodynamics from BL99 to MUSHY (Table 2) to examine whether
improved ice thermodynamics has noticeable influence on Arctic sea ice simulation and
prediction at seasonal timescale. Compared to Y21_RP, Y21_MUSHY (Fig. 4, pink line)
produces very similar evolution of the total ice extent. However, it simulates relatively larger
ice extent near the end of September, which is also reflected by the basin-wide increased ice
cover shown in Figure 5h$_3$. At the regional scale, compared to Y21_RP, Y21_MUSHY predicts
less ice in August in the Beaufort-Chukchi. The opposite is the case for the East Siberian-
Laptev Seas (Fig. 4a, b).
Figure 11 shows the difference of the ensemble mean of the predicted ice thickness
between Y21_MUSHY and Y21_RP. Compared with Y21_RP, Y21_MUSHY simulates
thicker ice (from ~0.2m in July to over 0.4m in September) extending from the Canadian Arctic,
through the central Arctic Ocean, to the Laptev Sea (Fig. 11a-c). This seems to be consistent
with previous studies, which show that the Mushy-layer thermodynamics simulates thicker ice
than BL99 thermodynamics in both standalone CICE (Turner and Hunke, 2015) and the fully-
coupled (Bailey et al., 2020), but Y21_MUSHY shows thinner ice (~0.2m) in an arc extending
from north of Alaska to north of eastern Siberia compared to Bailey et al. (2020). Note that
Y21_MUSHY focuses the effects of Mushy-thermodynamics on seasonal timescale while the
results in Bailey et al. (2020) are based on 50-year simulations.

Compared to Y21_RP, the mass budget of Y21_MUSHY (Fig. S6) shows that both surface

melting and frazil ice formation terms are increased. This compensation between surface
melting and frazil ice formation from the Mushy-layer thermodynamics in the CAPS leads to
relatively unchanged total ice extent between Y21_MUSHY and Y21_RP (Fig. 4 green and
pink lines).

**4.   Prediction skill of CAPS at longer timescale**
The design of Arctic sea ice prediction experiments described above follow the protocol
of the Sea Ice Prediction Network (SPIN), in which the outlook start from June 1[st], July 1[st], and
August 1[st] to predict seasonal minimum of the ice extent in September. It is not clear that how
predictive skills of dynamical models participating in SIPN may change for longer period. Here
we conduct two more experiments to investigate the predictive capability of CAPS beyond the
SPIN prediction period. For the prediction experiments discussed above, we use a simple
approach to merge CryoSat-2 and SMOS ice thickness by replacing ice thickness less than 1m
in CryoSat-2 data with SMOS data for ice thickness assimilation. Ricker et al. (2017) presented
a new ice thickness product (CS2SMOS) based on the optimal interpolation to statistically
merge CrySat-2 and SMOS data. Here we utilize the configuration of Y21_RP but use
CS2SMOS SIT for the assimilation (Y21_SIT; Table 2). The predicted total ice extent is almost
identical to Y21_RP in July but slightly larger total extent after July than that of Y21_RP (not
shown). The configuration of Y21_SIT is used in the following experiments. Taking advantage
of the entire prediction period provided by CFS forecasts (7 months), the Y21_EXT-7
experiment is designed to extend the prediction period to the end of January next year (Table
2). Figure 12 shows the temporal evolution of the ensemble mean of the predicted total Arctic
sea ice extent (as well as regional ice extent) for Y21_EXT-7. The total ice extent exhibits
reasonable evolution in terms of seasonal minimum and timing of recovery compared with the
observations until late November. Y21_EXT-7 also performs better than that of the
CLIM/DAMP predictions (black dashed and dotted lines) until mid-to-late November. After
that, Y21_EXT-7 overestimates the total ice extent relative to the observations, and such
overestimation is largely contributed by more extensive sea ice in the Barents-Kara-Greenland
Seas (Fig. 12c), which is a result of a sharp increase in the basal growth term after mid-to-late
November (not shown).

A growing number of studies have shown evidences of Arctic sea ice spring predictability

barrier. It means that predictions initialized prior to spring (before May) have much lower
predictive skill than predictions initialized after/on that date (e.g., Bonan et al., 2019; Bushuk
et al., 2017; 2018; Day et al., 2014). To investigate the predictive capability of CAPS initialized
prior to the summer melting season, the Y21_MAR-7 experiment is initialized on March 1$^{st}$,
2018 and predicts sea ice evolution until the end of September (Table 2). Figure 13 shows the
temporal evolution of the ensemble mean of the predicted total Arctic sea ice extent (as well as
regional ice extent) for the Y21_MAR-7 experiment. The evolution of predicted total sea ice
extent shows faster ice melting rate than the observations after mid-May, but slower ice
retreating after mid-July. As a result, the predicted minimum of ice extent has an overestimation
($\sim$1.2 million km$^2$) compared to the observed minimum. In contrast to Y21_MAR-7, the DAMP
prediction (black dotted line) agrees better with the observations throughout the 7-month
prediction period. At the regional scale, Y21_MAR-7 shows abrupt ice decline after May in the
Beaufort-Chukchi Seas (Fig. 13a), and this decline is mainly contributed by ice retreating along
the Alaskan coast (not shown). Sea ice in the East Siberian-Laptev Seas exhibits slow melting
after July (Fig. 13b), and ice cover still connect to the Siberian coast, which is different from
the observations (not shown). For the Barents-Kara-Greenland Seas (Baffin Bay-Canadian
Archipelago), there are systematic overestimations (underestimations) throughout the entire
prediction period (Fig. 13c-d). Bushuk et al. (2020) suggested that Arctic sea ice predictability
prior to the barrier date is mainly limited by synoptic events, which are only predictable for
few weeks, whereas the predictability after the barrier date is enhanced by ice-albedo feedback
with the onset of ice melting.

**5.    Conclusions and Discussions**

This paper presents and evaluates the updated Coupled Arctic Prediction System (CAPS)

designated for Arctic sea ice prediction through a case study for the year of 2018. A set of Pan-
Arctic prediction experiments with improved/changed physical parameterizations as well as
different configurations starting from July 1st to the end of September are performed for 2018
to assess their impacts of the updated CAPS on the predictive skill of Arctic sea ice at seasonal
timescale. Specifically, we focus on the Rapid Refresh (RAP) physics in the WRF model, the
oceanic tracer advection scheme in the ROMS model, sea ice thermodynamics in the CICE
model, and investigate physical process linking them to Arctic sea ice simulation and prediction.
The results show that the updated CAPS with improved physical parameterizations can
better predict the evolution of the total ice extent compared with its predecessor described in
Yang et al. (2020), though the predictions exhibit some prediction errors in regional ice extent.
The key improvements of WRF, including cumulus, boundary layer, and land surface schemes,
result in improved simulations in downward radiative fluxes and near surface air temperature.
These improvements mainly influence the predicted ice thickness instead of total ice extent.
The difference in the predicted ice thickness can have potential impacts on the icebreakers
planning their routes across the ice-covered regions. The major changes of ROMS, including
tracer advection and vertical mixing schemes, reduces the prediction error in sea surface
temperature and changes ocean temperature and salinity structure in the surface layer, leading
to improved evolution of the predicted total ice extent (particularly correcting the late ice
recovery issue in the previous CAPS). The change of CICE, including improved ice
thermodynamics, have noticeable influences on the predicted ice thickness.
We demonstrate that CAPS can remain skillful beyond the designated period of Sea Ice
Prediction Network (SIPN), which has potential values for stakeholders making decisions
regarding the socioeconomical activities. Although CAPS shows extended predictive skill to
the freeze-up period, the prediction produces extensive ice through the basal growth near the
end of prediction. The excessive basal growth may be partly due to that the bias of the CFS
data propagates into the model domain through lateral boundary conditions and its accumulated
effect influences Arctic sea ice simulation during the freeze-up period.
Keen et al. (2021) analyzed the Arctic mass budget of 15 models participated in the
Coupled Model Intercomparison Project Phase 6 (CMIP6). We notice that, first, the top melting
and the basal melting terms in CMIP6 models have comparable contributions in July while the
top melting term only has ~50% contribution relative to the basal melting term in the CAPS.
The updated CAPS with the RAP physics improves the performance of shortwave/longwave
radiation at the surface (Fig. 1 and Fig. 2). The net flux at the surface, however, may still be
underestimated in the CAPS. Besides, the surface property of sea ice (i.e., the amount of melt
ponds, bare ice, and snow) is a factor that influences surface albedo and thus the absorbed
shortwave radiation (e.g., Nicolaus et al., 2012; Nicolaus and Katlein, 2013). The prediction
experiments starting at July $1^{st}$ in this study do not consider the initialization of melt ponds (i.e.,
zero melt pond coverage at the initial). However, melt ponds start to develop in early May
based on the satellite observations (e.g., Liu et al., 2015, Fig. 1). The initialization of melt pond
based on the observations (e.g., Ding et al., 2020) in the CAPS is a direction to improve the
representation of the ice surface properties. Second, the mass budget analysis by both Keen et
al. (2021) and this study show that the contribution of lateral melting term is relatively small,
which might be due to that CMIP6 models and the CAPS assume constant floe-size (i.e., 300
meters in CICE), which is a critical value to determine the strength of lateral melting (e.g.,
Horvat et al., 2016; Steele, 1992). Recently, several studies have proposed floe size distribution
models (e.g., Bateson et al., 2020; Bennetts et al., 2017; Boutin et al., 2020; Horvat and
Tziperman, 2015; Roach et al., 2018, 2019; Zhang et al., 2015, 2016). Incorporating floe size
distribution model in the CAPS and understanding its impacts on seasonal Arctic sea ice
prediction will be a future direction of developing CAPS. Lastly, the prediction experiments
with the upwind advection scheme (i.e., Y21_RP, Y21_EXT-7) shows spurious large frazil ice
formation, particularity in July, which is different from the analysis shown in Keen et al. (2021).
An approach for reducing spurious frazil ice formation is proposed by Naughten et al. (2017)
that they implemented upwind limiter (Leonard and Mokhtari, 1990) to the U3H scheme to
further reduce the oscillations. Naughten et al. (2018) also suggested that the oscillatory
behaviors can be smoothed out by applying the Akima fourth-order tracer advection scheme
combined with Laplacian horizontal diffusion at a level strong enough. Beside of the oscillatory
behaviors of advection scheme, the ice-ocean heat flux can also play a role in the spurious frazil
ice formation. As discussed in section 3.3, the freezing/melting potential not only determines
the amount of newly-formed ice, but also limits the amount of energy that can be extracted
from the ocean surface layer to melt sea ice. This implies that the ocean surface layer will be
close to the freezing temperature if the ice-ocean heat fluxes reach the limit imposed by the
melting potential. Shi et al. (2021) discussed the impacts of different ice-ocean heat flux
parametrizations on sea ice simulations. Their results suggest that Arctic sea ice will be thicker
and ocean temperature will warmer beneath high-concentration ice with a complex approach
proposed by Schmidt et al. (2004) that limits melt rates (heat fluxes) of sea ice through
considering a fresh water layer underlying sea ice. The warmer ocean temperature under sea
ice with a more complex approach in ice-ocean heat flux may be the solution to reduce the
occurrence of local temperature falling below freezing temperature with oscillatory advection
schemes.
Based on the prediction experiments discussed in this paper, the configuration with the
RAP physics, the U3H/C4V ocean advection, BL99 ice thermodynamics, and CS2SMOS ice
thickness assimilation (Table 2, Y21_SIT) is assigned as the finalized CAPS version 1.0.
Improving the representation of physical processes in the CAPS version 1.0 for further
reducing the model bias will remain the main focus for the development of CAPS. Since the
CAPS is a regional modeling system, it relies on the forecasts form global climate models as
initial and lateral boundary conditions. That is, biases existed in GCM simulations (here the
CFS forecast) can be propagated into and affect the entire area-limited domain (e.g., Bruyère
et al., 2014; Rocheta et al., 2020; Wu et al., 2005). This issue can be a potential source that
influences the predictive capability of CAPS for longer timescales. Studies have applied bias
correction techniques with different complexities for improving the performance of regional
modeling system (e.g., Bruyère et al., 2014; Colette et al., 2012; Rocheta et al., 2017, 2020).
Further investigation is needed to address biases inherited from GCM predictions through
lateral boundaries for improving the predictive capability of CAPS.

Code and data availability: The COAWST and CICE models are open source and can be
downloaded from their developers at https://github.com/jcwarner-usgs/COAWST and
https://github.com/CICE-Consortium/CICE, respectively. PDAF can be obtained from
https://pdaf.awi.de/trac/wiki. CAPS v1.0 described in this paper is permanently archived at
https://doi.org/10.5281/zenodo.5034971. The prediction data analyzed in this paper can be
accessed from https://doi.org/10.5281/zenodo.4911415.

Author contributions: CYY and JL designed the model experiments, developed the
updated CAPS model, and wrote the manuscript, CYY conducted the prediction experiments
and analyzed the results. DC provided constructive feedback on the manuscript.

Competing interests: The authors declare that they have no conflict of interest.

Acknowledgements: This research is supported by the National Natural Science
Foundation of China (42006188), the National Key R&D Program of China
(2018YFA0605901), and the Innovation Group Project of Southern Marine Science and
Engineering Guangdong Laboratory (Zhuhai) (311021008). The authors also acknowledge the
National Centers for Environmental Prediction for providing CFS seasonal forecasts, the
University of Hamburg for distributing the SMOS sea ice thickness data, the Alfred-Wegener-
Institut, Helmholtz Zentrum für Polar- und Meeresforschung for providing the CryoSat-2 sea
ice thickness data and CS2SMOS data, the Polar Science Center for distributing the PIOMAS
ice thickness data, the National Snow and Ice Data Center for providing the SSMIS sea ice
concentration data, the European Centre for Medium-Range Weather Forecasts for distributing
the ERA5 reanalysis, and the National Oceanic and Atmospheric Administration for providing
the OI sea surface temperature.

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

Table 1 The summary of physic parameterizations used in the Y21_CRTL experiment

| WRF physics | |
| --- | --- |
| Cumulus parameterization | Grell-Freitas (Freitas et al. 2018; improved from Y20) |
| Microphysics parameterization | Morrison 2-moment (Morrison et al. 2009; same as Y20) |
| Longwave radiation parameterization | CAM spectral band scheme (Collins et al. 2004; same as Y20) |
| Shortwave radiation parameterization | CAM spectral band scheme (Collins et al. 2004; same as Y20) |
| Boundary layer physics | MYNN2 (Nakanishi and Niino, 2006; improved from Y20) |
| Land surface physics | Unified Noah LSM (Chen and Dudhia, 2001; improved from Y20) |
| | |
| ROMS physics | |
| Tracer advection scheme | MPDATA (Smolarkiewicz, 2006; same as Y20) |
| Tracer vertical mixing scheme | GLS (Umlauf and Burchard, 2003; same as Y20) |

| | |
|---|---|
| Bottom drag scheme | Quadratic bottom friction (QDRAG; (same as Y20) |
| | |
| CICE physics | |
| Ice dynamics | EVP (Hunke and Dukowicz, 1997; improved from Y20) |
| Ice thermodynamics | Bitz and Lipscomb (1999; same as Y20) |
| Shortwave albedo | Delta-Eddington (Briegleb and Light, 2007; same as Y20) |



Table 2 The summary of the prediction experiments and details of experiment designs.
Note: All experiments use the CFS operational forecasts as initial and boundary conditions; VT:
vertical transformation function; VS: vertical stretching function; SH94: stretching function of
Song and Haidvogel (1994); S10: stretching function of Shchepetkin (2010).

| Experiment | Physics | Assimilation | ROMS vertical coordinate | Simulation period |
|---|---|---|---|---|
| Y20_MOD | Physics (old version) listed in Table 1 | 6 localization radii SSMIS SIC Simply-merged CryoSat-2/SMOS SIT | VT 1 VS SH94 $h_c$ 10m | 2018.07.01- 2018.10.01 |
| Y21_CTRL | Physics (new version) listed in Table 1 | 6 localization radii SSMIS SIC Simply-merged CryoSat-2/SMOS SIT | VT 1 VS SH94 $h_c$ 10m | 2018.07.01- 2018.10.01 |
| Y21_VT | Physics (new version) listed in Table 1 | 6 localization radii SSMIS SIC Simply-merged CryoSat-2/SMOS SIT | VT 2 VS S10 $h_c$ 300m | 2018.07.01- 2018.10.01 |
| Y21_RP | Advection: U3H/C4V | 6 localization radii | VT 2 | 2018.07.01- |

| | | SSMIS SIC | VS S10 | 2018.10.01 |
| | | Simply-merged CryoSat-2/SMOS SIT | $h_c$ 300m | |
| Y21_MUSHY | Same physics as Y21_RP | 6 localization radii | VT 2 | 2018.07.01- |
| | CICE: Mushy layer thermodynamics | SSMIS SIC | VS S10 | 2018.10.01 |
| | | Simply-merged CryoSat-2/SMOS SIT | $h_c$ 300m | |
| Y21_ SIT | Same physics as Y21_RP | 6 localization radii | VT 2 | 2018.07.01- |
| | | SSMIS SIC | VS S10 | 2018.10.01 |
| | | OI-merged CryoSat-2/SMOS SIT | $h_c$ 300m | |
| Y21_EXT-7 | Same physics as Y21_RP | 6 localization radii | VT 2 | 2018.07.01- |
| | | SSMIS SIC | VS S10 | 2019.01.31 |
| | | OI-merged CryoSat-2/SMOS SIT | $h_c$ 300m | |
| Y21_MAR-7 | Same physics as Y21_RP | 6 localization radii | VT 2 | 2018.03.01- |
| | | SSMIS SIC | VS S10 | 2018.09.30 |
| | | OI-merged CryoSat-2/SMOS SIT | $h_c$ 300m | |



    **8. Figures**

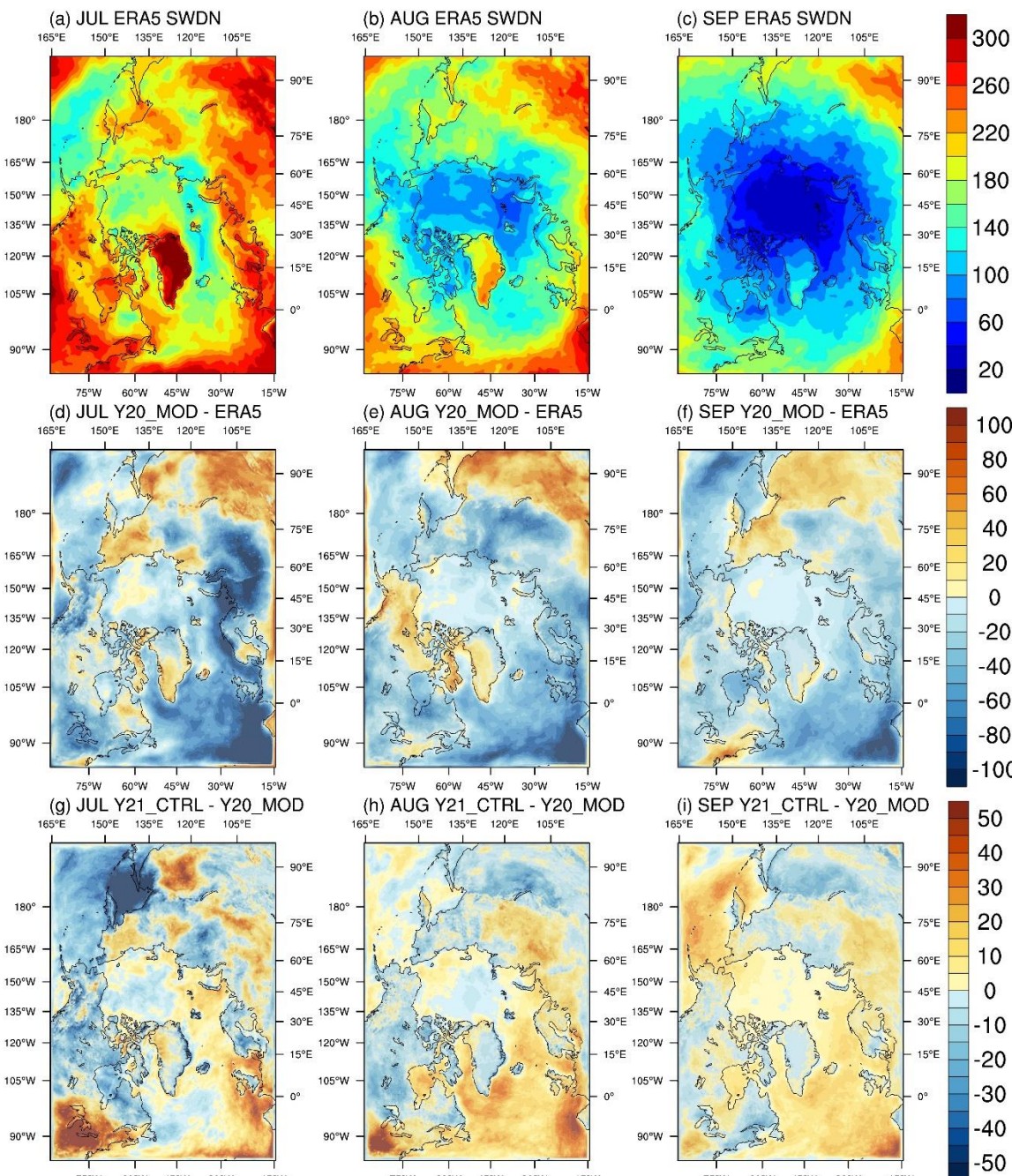


Figure 1 ERA5 monthly mean of downward shortwave radiation at the surface for (a) July, (b)

August, and (c) September, the difference between Y20_MOD and ERA5 for (d) July, (e)

August, (f) September, and the difference between Y21_CTRL and Y20_MOD for (g) July, (h)

August, and (i) September.


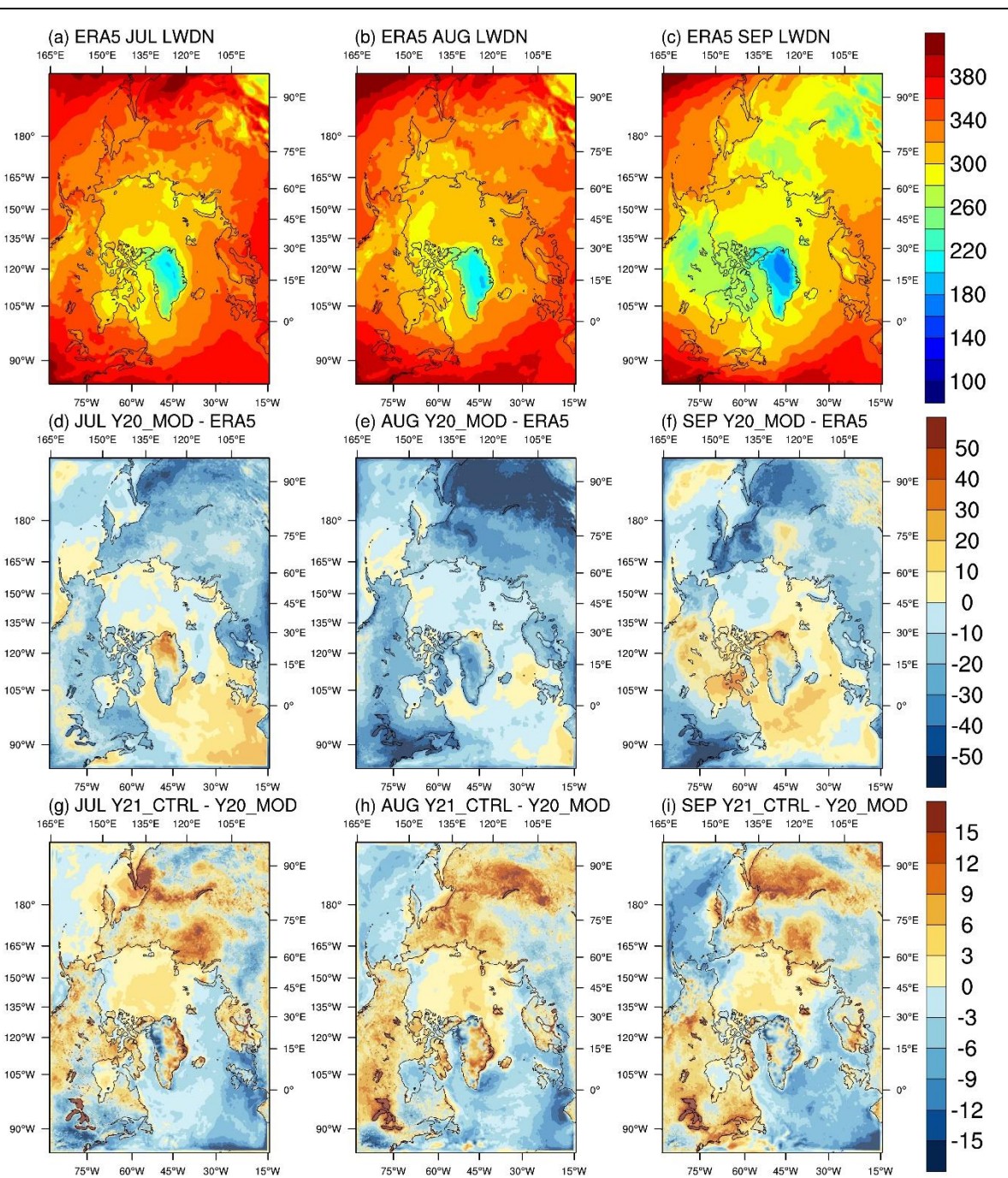

Figure 2 Same as Figure 1, but for downward thermal radiation at the surface.

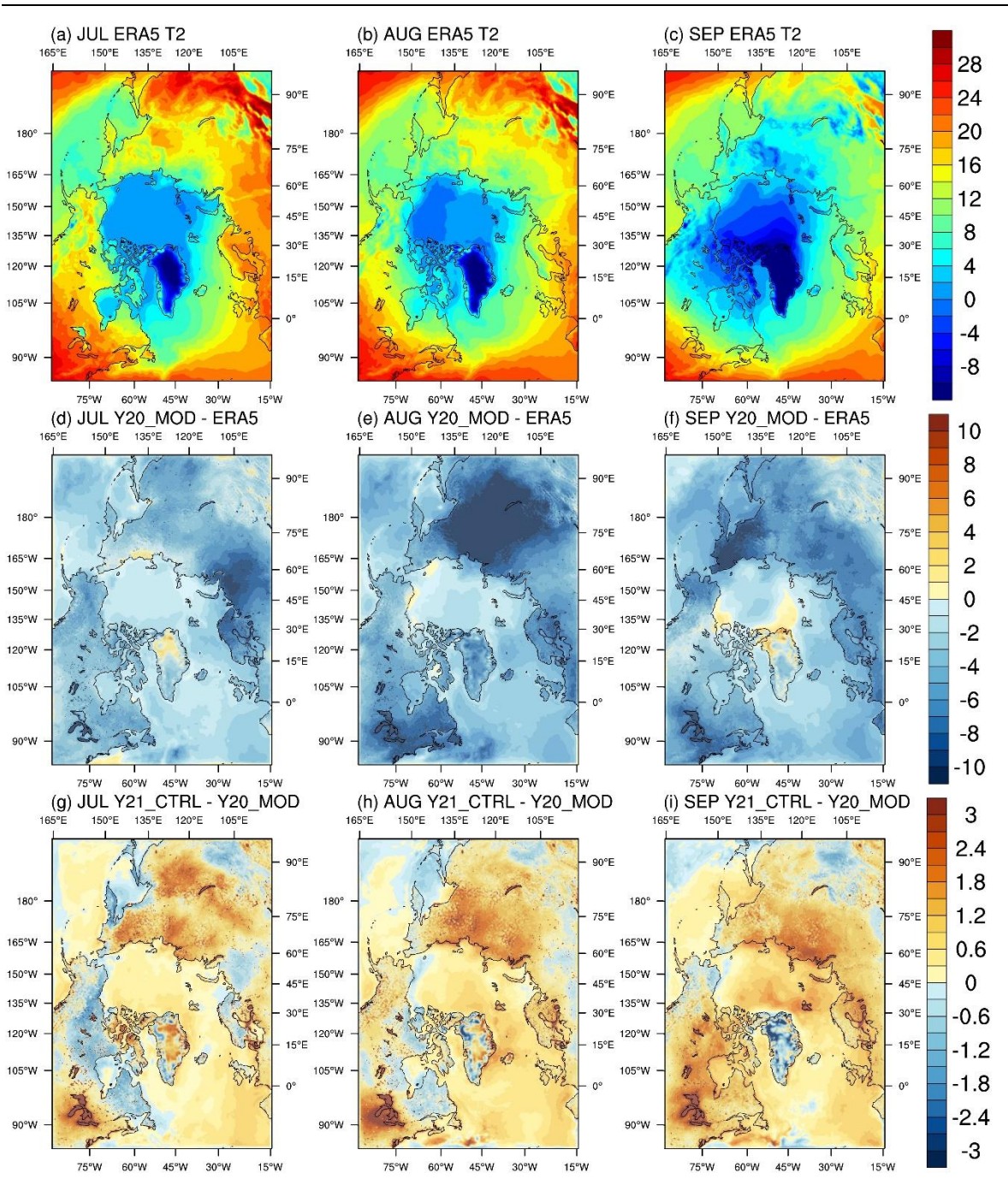

Figure 3 Same as Figure 1, but for near-surface air temperature.


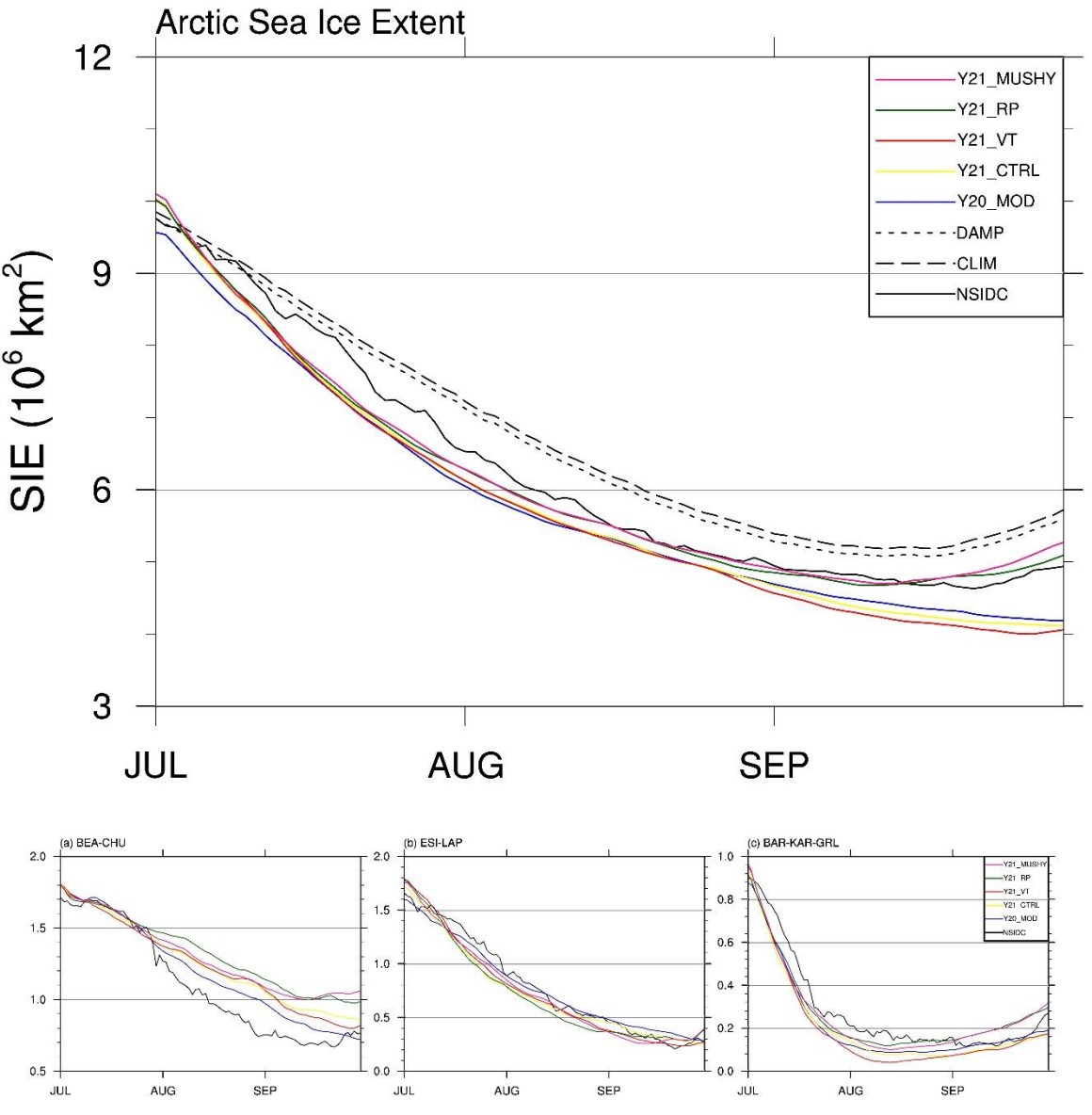


Figure 4 Top panel: Time-series of Arctic sea ice extent for the observations (black line) and
the ensemble-mean of Y20_MOD (blue line), Y21_CTRL (yellow line), Y21_VT (red line),
Y21_RP (green line), and Y21_MUSHY (pink line). Dashed and dotted lines are the
climatology and the damped anomaly persistence predictions. Bottom panel: Time-series of
the observed (black line) and the ensemble-mean of regional sea ice extents for Y20_MOD
(blue line), Y21_CTRL (yellow line), Y21_VT (red line), Y21_RP (green line), and
Y21_MUSHY (pink line) for (a) Beaufort-Chukchi Seas, (b) East Siberian-Laptev Seas, and
(c) Barents-Kara-Greenland Seas.

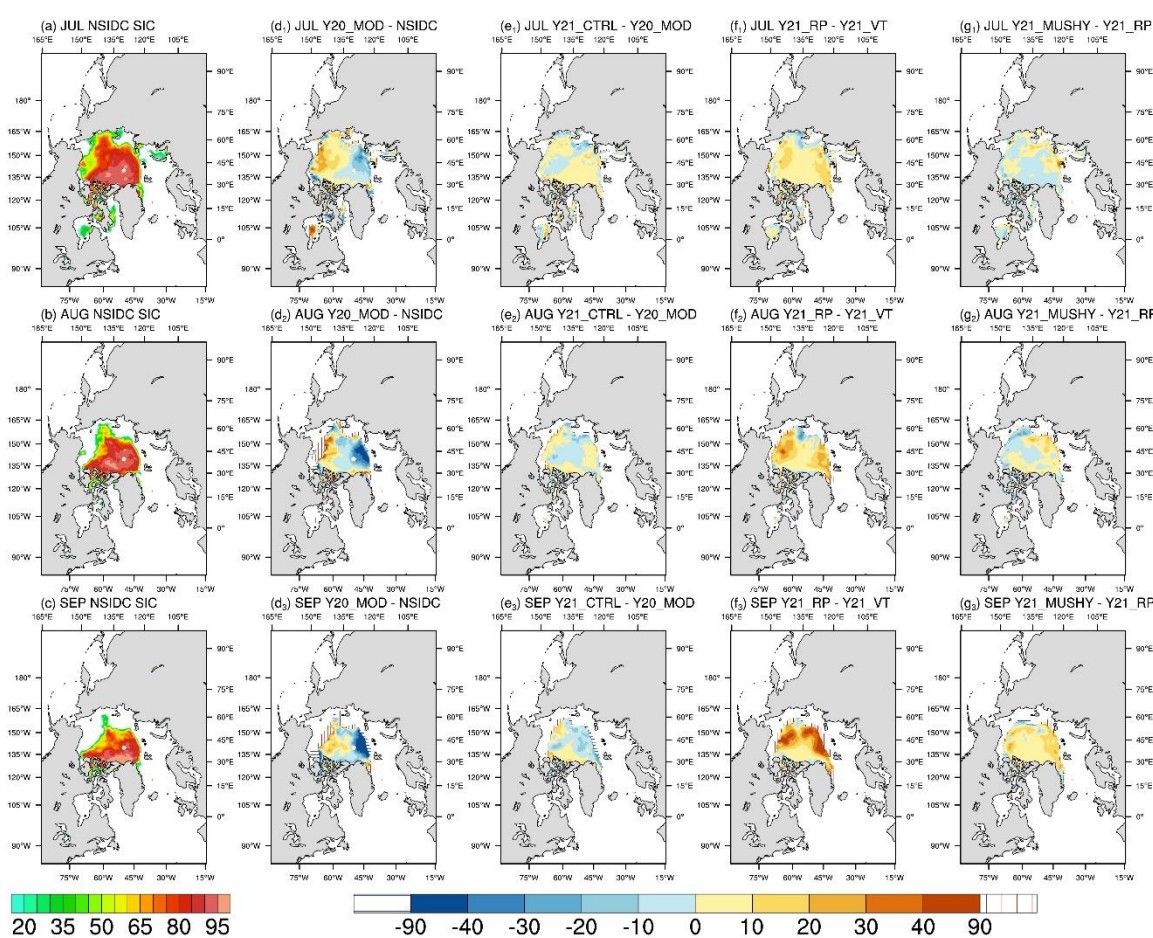


Figure 5 Monthly mean of sea ice concentration for (a) July, (b) August, (c) September of the
NSIDC observations, and the difference between the all prediction experiments and the
observations for (d$_1$-g$_1$) July, (d$_2$-g$_2$) August, (d$_3$-g$_3$) September. Vertical/horizontal-line areas
represent the difference of ice edge location (15% concentration).


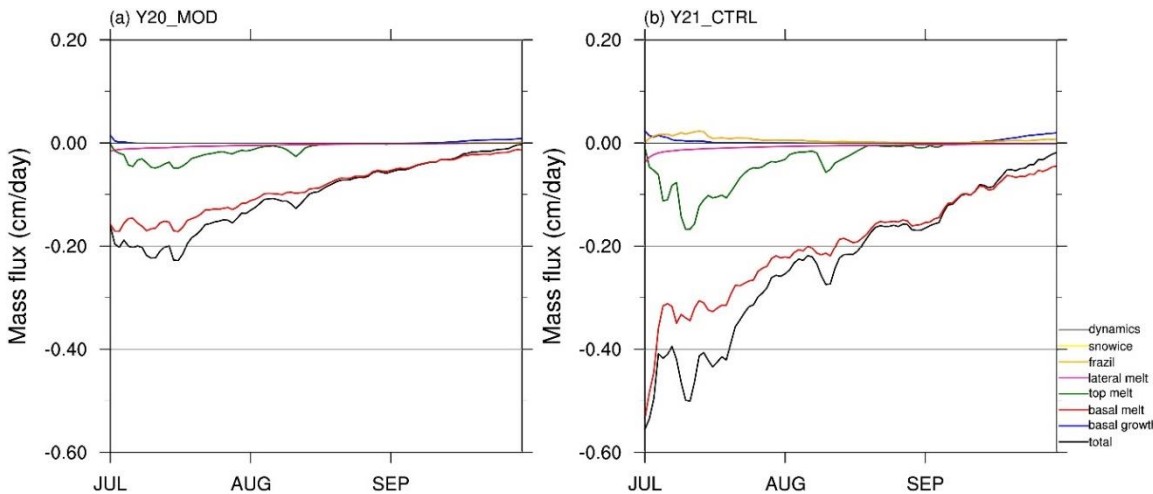


Figure 6 Time-series of sea ice mass budget terms for (a) Y20_MOD and (b) Y21_CTRL.


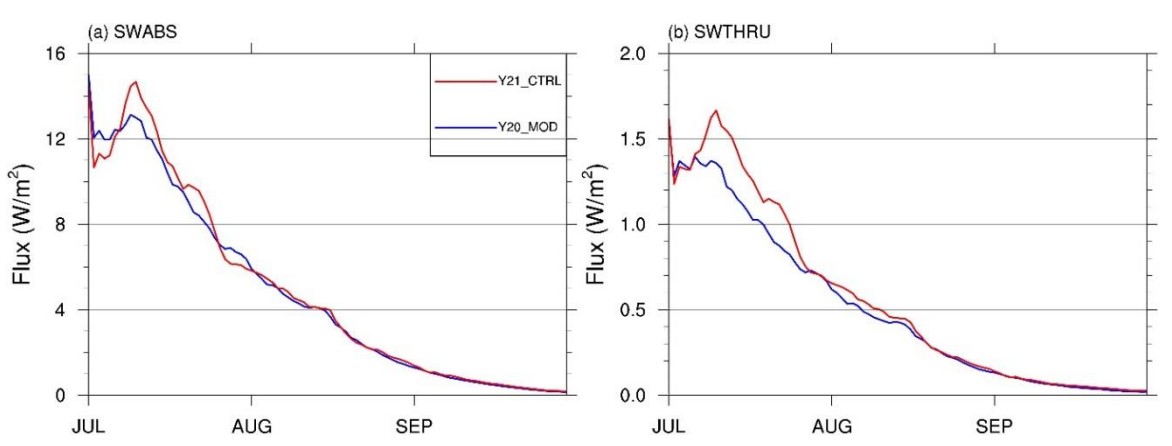


Figure 7 Time-series of (a) shortwave radiation absorbed by ice surface, and (b) penetrating
shortwave radiation to the upper ocean averaged over ice-covered grid cells for Y20_MOD
(blue line) and Y21_CTRL (red line).

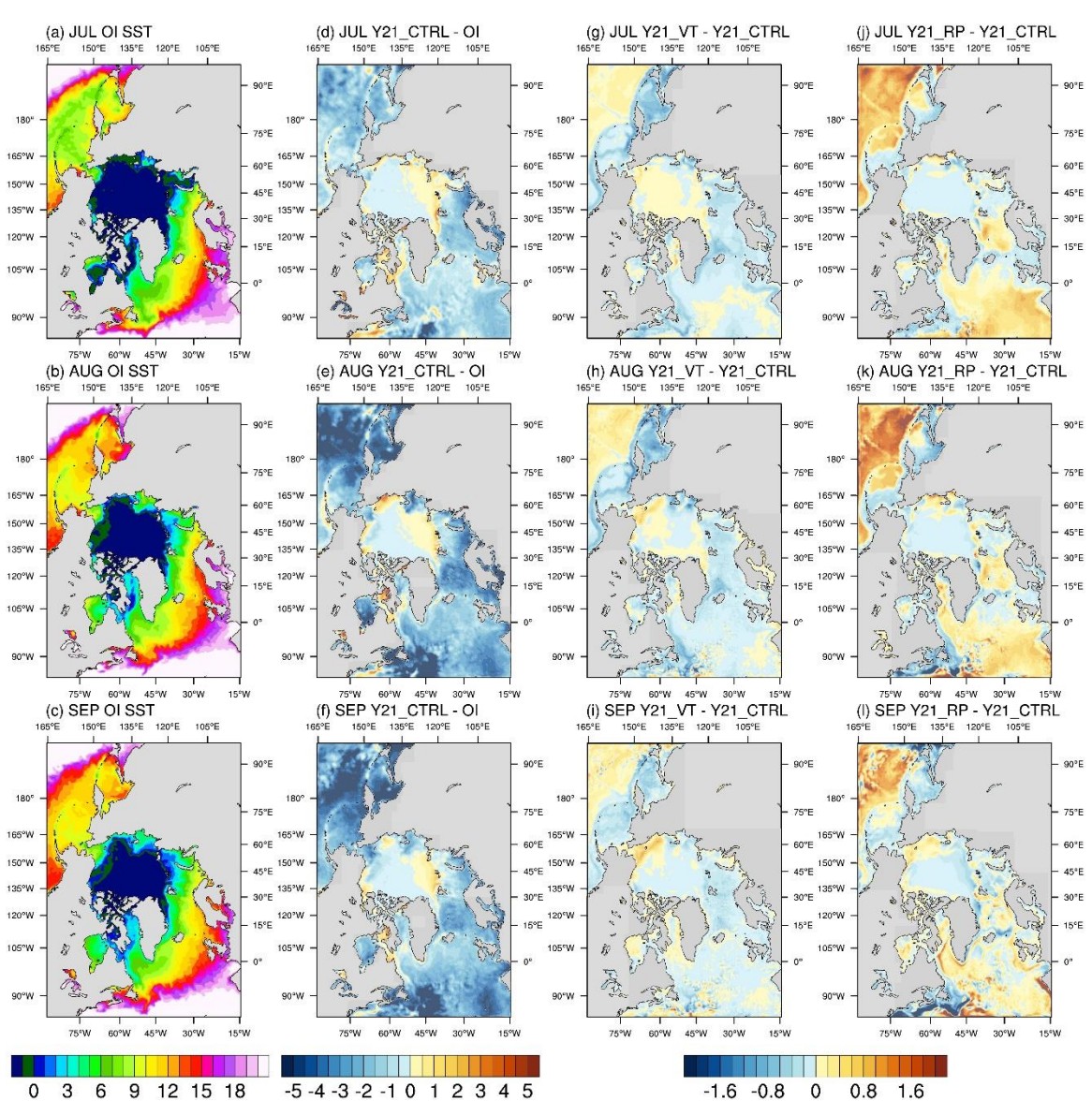

Figure 8 First column: monthly mean of sea surface temperature for (a) July, (b) August, (c) September of the OI SST. Second column: the difference between Y21_CTRL and the OI SST for (d) July, (e) August, (f) September. Right panel: Monthly mean of sea surface temperature difference between Y21_VT/Y21_RP and Y21_CTRL for (g) July, (h) August, (i) September of Y21_VT, (j) July, (k) August, and (l) September of Y21_RP.

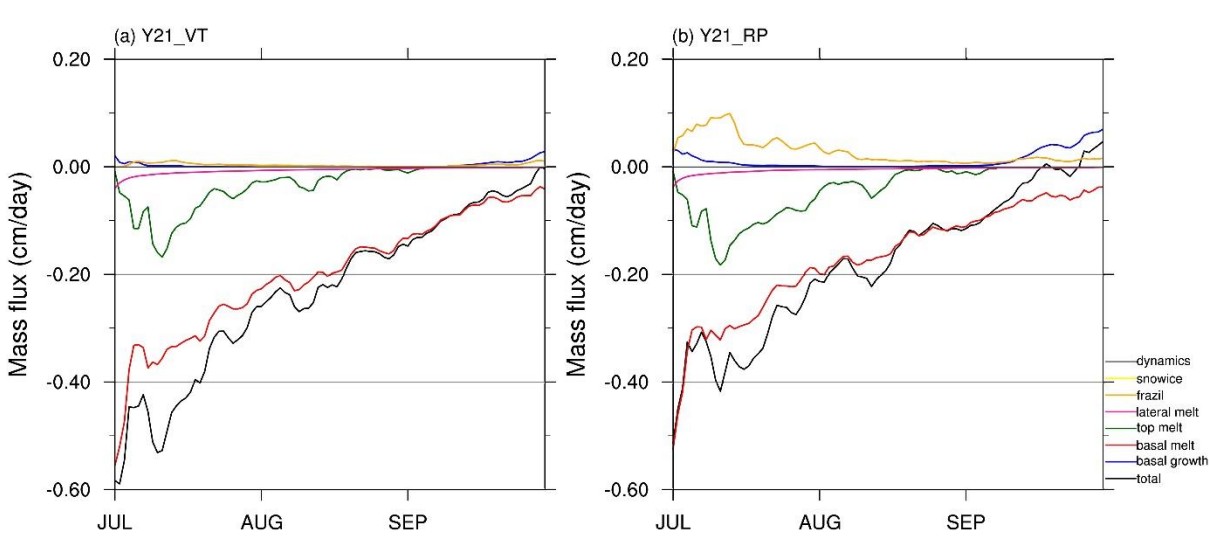

Figure 9 Same as Figure 6, but for (a) Y21_VT, and (b) Y21_RP.

1006

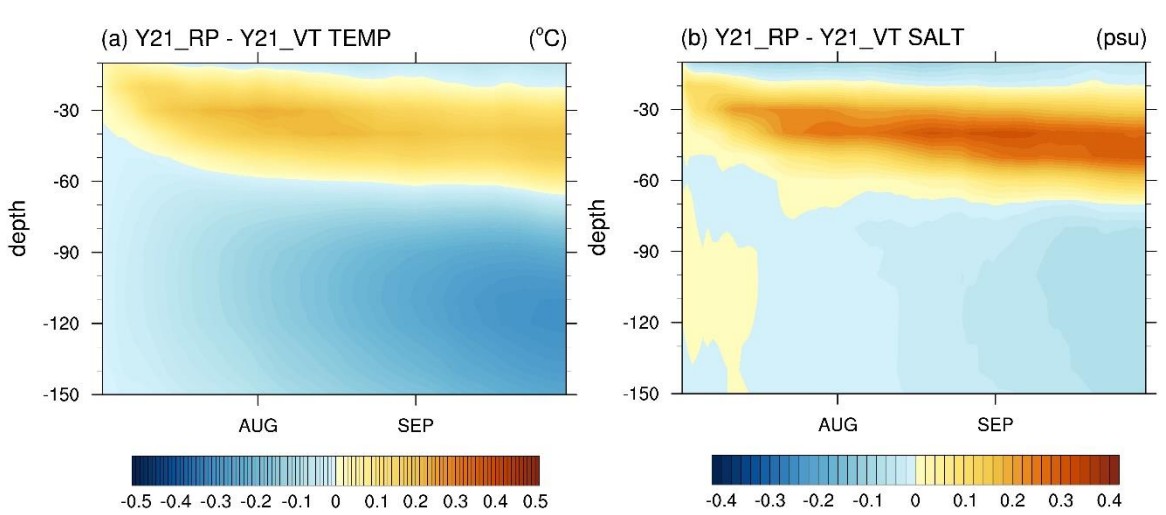

Figure 10 (a) the average temperature profile of upper 150 m under ice-covered areas for the difference between Y21_RP and Y21_VT. (b) same as (a), but for the salinity profile.

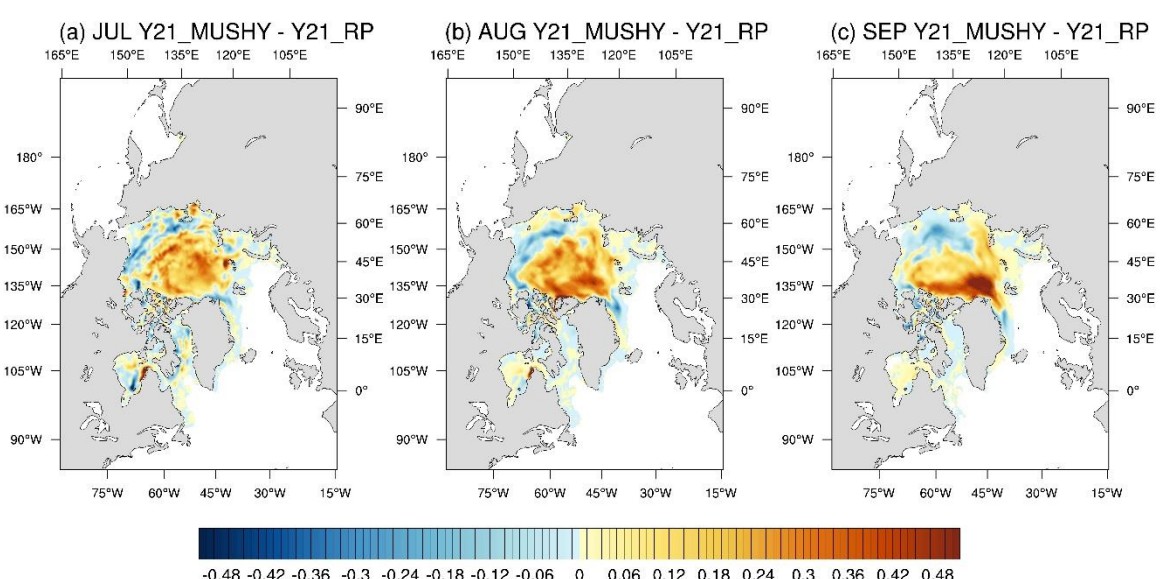

Figure 11 Monthly mean of sea ice thickness difference between Y21_MUSHY and Y21_RP
for (a) July, (b) August, and (c) September.

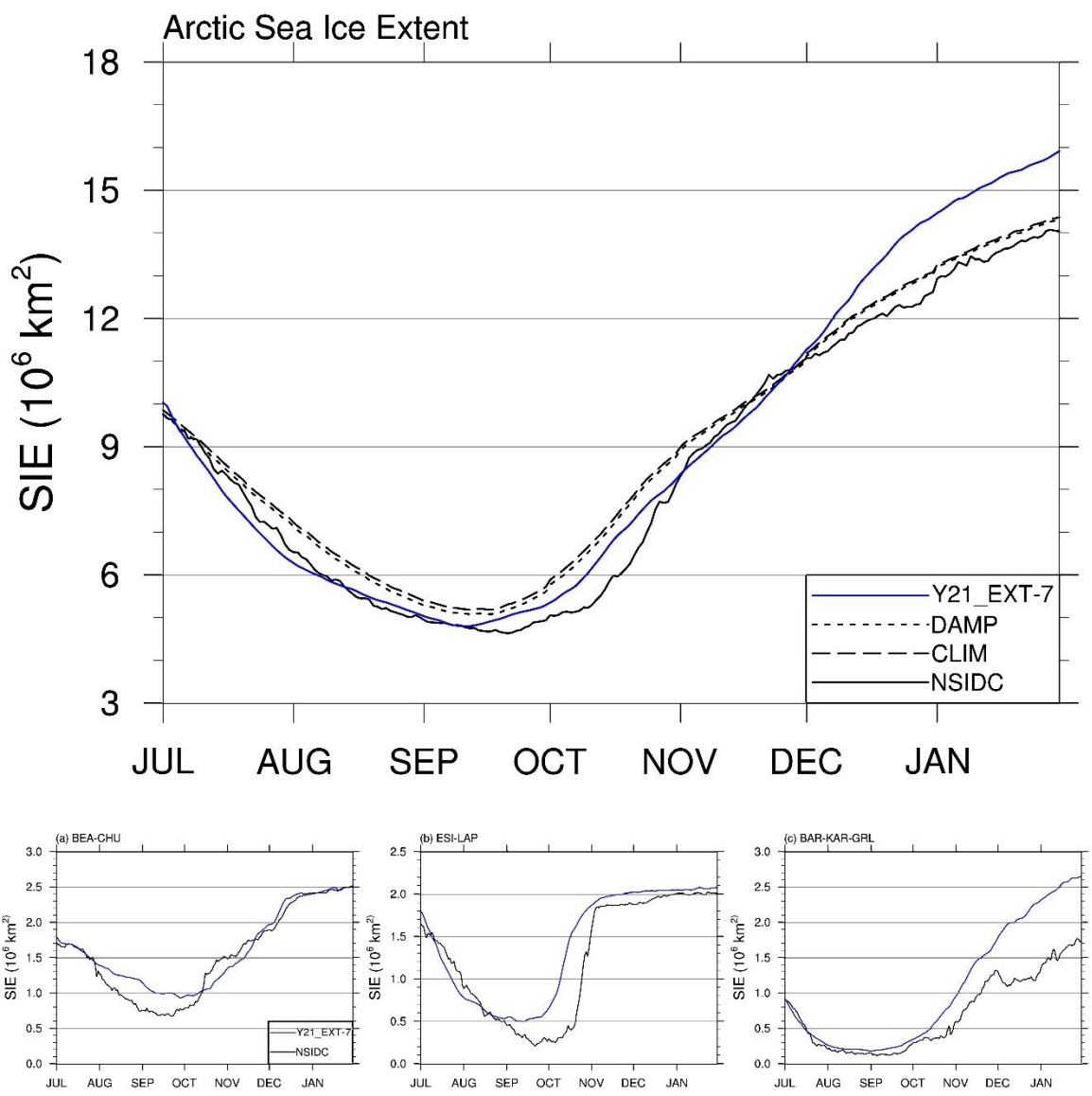


Figure 12 Same as Figure 4, but for Y21_EXT-7.

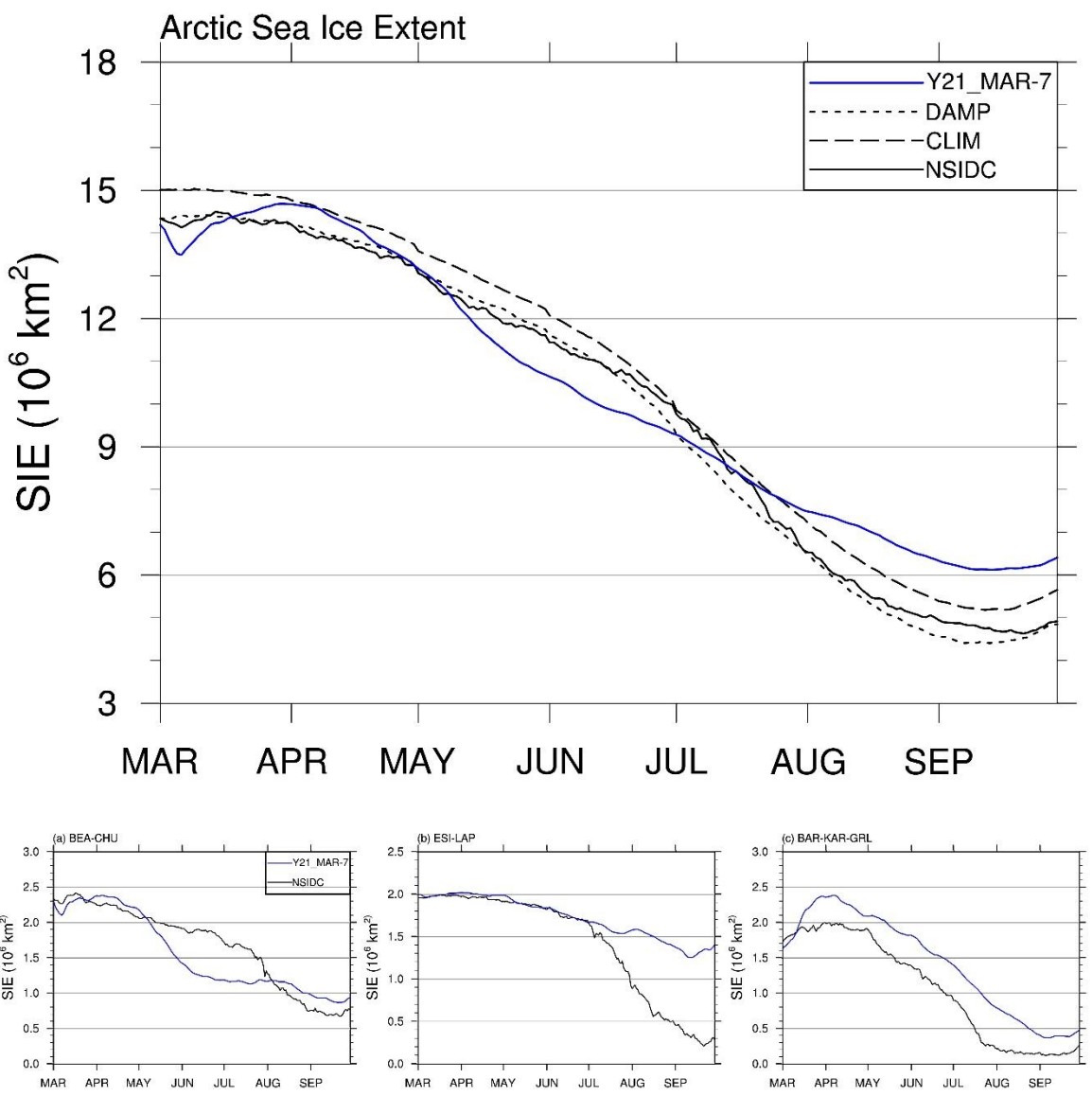


Figure 13 Same as Figure 4, bur for Y21_MAR-7.