# Peer review of "An improved regional coupled modeling system for Arctic sea ice simulation and"

_Geoscientific Model Development, 2021_

## Author Comment (AC1)

**Response to the reviews of** "CAPS v1.0: An improved regional coupled modeling system for Arctic sea ice and climate simulation and prediction" by Chao-Yuan Yang, Jiping Liu, Dake Chen

**Now the title is changed to "An improved regional coupled modeling system for Arctic sea ice simulation and prediction: a case study for 2018"**

**General response to all reviewers**

**Based on the reviewers' general comments, we have made substantial changes to the manuscript. In the revision, we focused on four scientific questions that we want to address with the improved/updated Coupled Arctic Prediction System in the introduction, including:**

**1) to what extent the improved convection and boundary layer schemes in the atmospheric model can improve atmospheric simulations in the Arctic (i.e., radiation, temperature, humidity, and wind), and then benefit seasonal Arctic sea ice simulation and prediction;**

**2) to what extent different advection schemes in the oceanic model can change the simulation of upper ocean structure, and then influence Arctic sea ice prediction;**

**3) whether the more realistic sea ice thermodynamics scheme can produce noticeable influence on seasonal Arctic sea ice prediction;**

**4) whether the updated Arctic prediction system has predictive skill for longer periods.**

**To address the issue that the original manuscript did not offer sufficient analyses of physical process linking improved/changed physical parameterizations to simulated sea ice state, in the revision, we added more and in-depth analyses, particularly sea ice mass budget analysis that is used to separate sea ice mass changes by different physical processes for all experiments, including 1) sea ice growth in supercooled open water (frazil ice formation), 2) basal growth, 3) snow-to-ice conversion, 4) top melt, 5) basal melt, 6) lateral melt, and 7) dynamics process (proposed by Notz et al., 2016). In order to perform the mass budget analysis, we re-conducted all model experiments and outputted the related variables to calculate sea ice mass budget. Built on the insight from the comparison of sea ice mass budget between different experiments, we show specific processes linking improved/changed physical parameterizations to simulated**

sea ice state, i.e., changes in shortwave radiation absorbed by ice surface and penetrating shortwave radiation to the upper ocean induced by the improved convection and boundary layer schemes in the atmospheric model component; changes in vertical profiles of the upper ocean induced by different advection schemes in the ocean model component.

Note: We made the point-to-point response to the reviewer's comments (see below for details). Since we have made substantial changes to the manuscript, it is difficult to put all changes in the response letter. Here we provide a web link here for the reviewers accessing the revised manuscript with all highlighted changes (https://doi.org/10.5281/zenodo.5739272).

*Reference: Notz, D., Jahn, A., Holland, M., Hunke, E., Massonnet, F., Stroeve, J., Tremblay, B., and Vancoppenolle, M.: The CMIP6 Sea-Ice Model Intercomparison Project (SIMIP): understanding sea ice through climate-model simulations, Geosci. Model Dev., 9, 3427–3446, https://doi.org/10.5194/gmd-9-3427-2016, 2016.*

**Response to comments by Reviewer #1**

**We would like to thank the reviewer for the helpful comments on the paper.**

General comments:

The authors evaluate the updated version of Arctic prediction system in predicting summer Arctic sea ice in 2018. The prediction system CAPSv1.0 consists of model components for atmosphere, ocean and sea ice, and the sea ice parameters are initialized by assimilating observational information. The prediction system CAPS has proven some skill in predicting summer Arctic sea ice in earlier works, such as Yang et al (2020). Changing only one set of parameterizations (or configurations) in one model component allows to study the origin of improved prediction skill in different Arctic regions. However, there is lack of novelty in the analysis by repeating the results from a number of experiments and calculating the differences between them and a reference experiment (Yang et al, 2020), with key parameters of sea ice extent and sea ice concentration. Due to the number of performed experiments and the material to be discussed, highlight of the main findings is unclear and the major benefit from improved physics is not addressed. The model evaluation is mainly based on a case study of Arctic sea ice in summer 2018, the year with sixth lowest summertime minimum

extent in the satellite record. The authors attempt to predict the slow recovery of sea ice cover in the Chukchi and Barents Sea, by extending the seasonal forecast over autumn. Although there is mismatch of the timing of reforming sea ice in the concerned regions, potential modelling solution or missing physics is not discussed.

My primary concern is that the paper has not yet met the GMD publication standards for model description papers for the following reasons:

1) authors should provide more technical details about the improved physical features than referring to several publications

**Response: Thanks for the reviewer's suggestion. In this revision, we added more technical details about the improved physical parameterizations. Using the improvement in the atmospheric model component (WRF) as an example, the Rapid Refresh (RAP) model has made some improvements in the WRF model physics (Benjamin et al., 2016), including improved Grell-Freitas convection scheme (GF) and Mellor-Yamada-Nakanishi-Niino planetary boundary layer scheme (MYNN). For the GF scheme, the major improvements relative to the original scheme (Grell and Freitas, 2014) include:**

**1) a beta probability density function used as the normalized mass flux profile for representing height-dependent entrainment/detrainment rates within statistical-averaged deep convective plumes, which is given as:**

$$Z_{u,d}(r_k) = cr_k^\alpha - (1 - r_k)^\beta - 1$$

**where $Z_{u,d}$ is the mass flux profiles for updrafts and downdrafts, c is a normalization constant, $r_k$ is the location of the mass flux maximum, $\alpha$ and $\beta$ determine the skewness of the beta probability density function**

**2) the ECMWF approach used for momentum transport due to convection (Biswas et al. 2020; Freitas et al. 2018; 2021). For the MYNN scheme, the RAP model improves the mixing-length formulation, which is designed as:**

$$\frac{1}{l_m} = \frac{1}{l_s} + \frac{1}{l_t} + \frac{1}{l_b}$$

**where $l_m$ is the mixing length, $l_s$ is the surface length, $l_t$ is the turbulent length, and $l_b$ is the buoyancy length. Compared to the original scheme, the RAP model changed coefficients in the formulation of $l_s$, $l_t$, and $l_b$ for reducing the near-surface turbulent**

mixing, and the diffusivity of the scheme. The RAP model also and removes numerical deficiencies to better represent subgrid-scale cloudiness (Benjamin et al. 2016, see Append. B) compared to the original scheme (Nakanishi and Nino, 2009).

*Reference:*

*Biswas, M. K., Zhang, J. A., Grell, E., Kalina, E., Newman, K., Bernardet, L., Carson, L., Frimel, J., and Grell, G.: Evaluation of the Grell–Freitas Convective Scheme in the Hurricane Weather Research and Forecasting (HWRF) Model, Weather and Forecasting, 35(3), 1017-1033, 2020.*

*Benjamin, S. G., Weygandt, S. S., Brown, J. M., Hu, M., Alexander, C. R., Smirnova, T. G. and Manikin, G. S.: A North American hourly assimilation and model forecast cycle: the Rapid Refresh. Monthly Weather Review, 144,1669–1694. https://doi.org/10.1175/MWR-D-15-0242.1, 2016.*

*Freitas, S. R., Grell, G. A., Molod, A., Thompson, M. A., Putman, W. M., Santos e Silva, C. M. and Souza, E. P.: Assessing the Grell–Freitas convection parameterization in the NASA GEOS modeling system. J. Adv. Model. Earth Syst., 10, 1266–1289, https://doi.org/10.1029/2017MS001251, 2018.*

*Freitas, S. R., Grell, G. A., and Li, H.: The Grell–Freitas (GF) convection parameterization: recent developments, extensions, and applications, Geosci. Model Dev., 14, 5393–5411, https://doi.org/10.5194/gmd-14-5393-2021, 2021.*

We also added more technical details about the improved/changed physical parameterizations in other experiments and the revised texts are referred to the revised manuscript.

2) authors should provide more materials and highlight the major benefit of physical changes in reducing prediction errors in seasonal forecast of Arctic sea ice.

Response: We thank the reviewer for the helpful comment. Based on the comment, in this revision, we added sea ice mass budget analysis to all experiments and analyzed the contributing process to the changes of sea ice. Using the improvement in the atmospheric model component (WRF) as an example, Figure R1 shows the evolution of sea ice mass budget terms of Y20_MOD and Y21_CTRL. Compared with Y20_MOD (Fig. R1a), Y21_CTRL (Fig. R1b) shows much larger magnitude for basal and surface

melt. In a fully coupled predictive model, the changes of sea ice are determined by the fluxes from the atmosphere above and the ocean below. Associated with the increased downward radiation induced by the RAP physics, Y21_CTRL absorbs more shortwave radiation (SWABS, Fig. R2a) and allows more penetrating solar radiation into the upper ocean below sea ice (SWTHRU, Fig. R2b) than that of Y20_MOD, especially in July. This explains why Y21_CTRL has larger magnitude of surface and basal melting terms. As a result, Y21_CTRL produces thinner ice thickness than that of Y20_MOD, in the East Siberian-Laptev Seas in July and in the much of central Arctic Ocean in August and September (Fig. R3). We also conducted similar analyses for other experiments and the revised texts are referred to the revised manuscript.

[Figure]

**Figure R1 Time-series of sea ice mass budget terms for (a) Y20_MOD and (b) Y21_CTRL.**

[Figure]

**Figure R2 Time-series of (a) shortwave radiation absorbed by ice surface, and (b) penetrating shortwave radiation to the upper ocean averaged over ice-covered grid cells for Y20_MOD (blue line) and Y21_CTRL (red line).**

[Figure]

**Figure R3 Monthly mean of sea ice thickness difference for (a) July, (b) August, and (c) September between Y21_CTRL and Y20_MOD.**

3) Regarding the evaluation of the Arctic prediction system, the present analysis based on only one year case study is insufficient to perform skill assessment. The authors should introduce novel ways of comparing model results with observational data.

**Response: We agree with the reviewer that the predictive skill assessment with a suite of hindcasts across multiple years is essential. The primary objective of this study is to understand how improved/changed physical parameterizations in the updated Coupled Arctic Prediction System influence Arctic sea ice simulation and prediction compared to its predecessor described in Y20 by focusing on physical process linking improved/changed physical parameterizations to simulated Arctic sea ice. Thus in this study, we conducted Arctic sea ice simulation and prediction for the year of 2018 as a case study. As suggested the reviewer, we changed the title to "An improved regional coupled modeling system for Arctic sea ice simulation and prediction: a case study for 2018" and revised the main text to reflect this. Additionally, the predicted total sea ice extent of all experiments are compared with the common benchmarks, including the climatology prediction (CLIM) and the damped anomaly persistence prediction (DAMP) which are also used in other studies (e.g., Blanchard-Wriggleworth et al., 2015; Wayand et al., 2019). The skill assessment with a suite of hindcasts across multiple years will be performed with the finalized version of the prediction system in the future study.**

*Blanchard-Wrigglesworth, E., Cullather, R. I., Wang, W., Zhang, J., and Bitz, C. M.:*

*Model forecast skill and sensitivity to initial conditions in the seasonal Sea Ice Outlook, Geophys. Res. Lett., 42, doi:10.1002/2015GL065860, 2015.*

*Wayand, N. E., Bitz, C. M., and Blanchard-Wrigglesworth, E.: A year-round subseasonal-to-seasonal sea ice prediction portal. Geophysical Research Letters, 46, 3298–3307. https://doi.org/10.1029/2018GL081565, 2019.*

4) As the authors primarily discuss on the melting season of Arctic sea ice in 2018 and most experimental runs are no more than three months long, a little adjustment of the title may be helpful, such as by replacing "and climate simulation and prediction" by "in summer 2018".

**Thanks for the reviewer's comment. We adjusted the title to "An improved regional coupled modeling system for Arctic sea ice simulation and prediction: a case study for 2018".**

Concerning the amount of work to be done to reach the standard of publication with GMD, I suggest the authors consider resubmission after substantial improvement.

Specific major comments:

1. Better visualization for experiment design

F.ex. Table 3, please consider to put repeated parameterization (or configuration) to identical color. Figures of Arctic sea ice extent time series, e.g. Fig2, 3, 8, 11, 13. The line color could be consistent with the color used in Table 3 to distinguish from different model components. For the spatial maps of sea ice parameters, you should add some text in big front in figure or figure caption to highlight the improved model components.

**Response: Based on the reviewer's suggestion, we modified the line color so the figures have consistent color for repeated configurations, and added texts in figure captions. For example, we replotted time series of Arctic sea ice extent in a single figure (Figure R4). This is also true for spatial distribution of Arctic sea ice concentration (Figure R5).**

[Figure]

**Figure R4 Top panel: Time-series of Arctic sea ice extent for the observations (black line) and the ensemble-mean of Y20_MOD (blue line), Y21_CTRL (yellow line), Y21_VT (red line), Y21_RP (green line), and Y21_MUSHY (pink line). Dashed and dotted lines are the climatology and the damped anomaly persistence predictions. Bottom panel: Time-series of the observed (black line) and the ensemble-mean of regional sea ice extents for Y20_MOD (blue line), Y21_CTRL (yellow line), Y21_VT (red line), Y21_RP (green line), and Y21_MUSHY (pink line) for (a) Beaufort-Chukchi Seas, (b) East Siberian-Laptev Seas, and (c) Barents-Kara-Greenland Seas.**

[Figure]

**Figure R5** Monthly mean of sea ice concentration for (a) July, (b) August, (c) September of the NSIDC observations, and the difference between the all prediction experiments and the observations for (d₁-g₁) July, (d₂-g₂) August, (d₃-g₃) September. Vertical/horizontal-line areas represent the difference of ice edge location (15% concentration).

2. Assessment of the results

I think the existing figures of the results contain very basic information and could be moved to supplementary. Main results should invovle more comprehensive statistical analysis and present more concrete evidence of improvements.

**Response: Based on the reviewer's suggestion. In this revision, we added a supplementary material. We put the results contain very basic information (i.e., no significant change) to the supplementary. In the main body, we focus on the analyses showing more concrete evidence of physical process linking improved/changed physical parameterizations to simulated sea ice state, i.e., built on the insight from the**

**comparison of sea ice mass budget between different experiments, we showed specific processes linking changes in sea ice simulations.**

3. Identification of improved key process in sea ice seasonal prediction

- Wind-driven ocean currents and sea ice export has been identified as key factors in the retreat of Arctic sea ice during summer in the Beaufort Gyre (Armitage et al, 2020) and Barents Sea (Dai et al, 2020). The authors may have a close look at the wind anomalies from the changed atmopheric model.

**Response: Thanks for the reviewer's comment. Regarding to improved convection and boundary layer schemes in the atmospheric model, we further analyzed the 10-meter winds for Y20_MOD and Y21_CTRL that are directly influenced by the changed atmospheric model. The results show that both Y20_MOD and Y21_CTRL have similar circulation patterns as demonstrated in Figure R6. This suggests that the simulated Arctic sea ice changes are mainly induced by the thermodynamic effect of the improved convection and boundary layer schemes in the atmospheric model. That is the change in surface radiative fluxes (detailed discussion can be found in the revised manuscript)**

[Figure]

**Figure R6 ERA5 monthly mean of 10-meter winds for (a) July, (b) August, and (c) September, (d) July, (e) August, (f) September of Y20_MOD, and (g) July, (h) August, and (i) September of Y21_CTRL.**

- With regard to improved configuration in the ocean model, it would be scientifically interesting to investigate which factor plays a dominant role in sea ice melting, e.g. changes in sea surface temperature, salinity or eddy activity? SST can be the driver, but also the effect in the coupled system with changed physics. It would be more convencing by showing vertical profiles of ocean temperature and salinity in the concerned region.

**Response: Based on the reviewer's suggestion. First, we conducted sea ice mass budget analysis. Figure R7 shows the evolution of sea ice mass budget terms of Y21_VT and Y21_RP. Relative to Y21_VT, Y21_RP (with U3H/C4V scheme) results in increased frazil ice formation in July, which is partly compensated by increased surface melting. Y21_RP also leads to increased basal growth in mid- and late September (Fig. R7a, b). Second, we investigated the factor (vertical structure of ocean temperature and salinity) responsible for the changes from sea ice mass budget analysis. Figure R8 shows the difference in the vertical profile of ocean temperature and salinity in the upper 150 m averaged for the central Arctic Ocean between Y21_RP and Y21_VT. The ocean temperature in the surface layer of Y21_RP is slightly colder during the prediction period compared to that of Y21_VT (Fig. R8a), especially in August and September. Moreover, the water in the surface layer (0-20 m) of Y21_RP is fresher than that of Y21_VT (Fig. R8b). They reduce the freezing temperature and favor frazil ice formation. In the CAPS, the frazil ice formation is determined by the freezing potential, which is the vertical integral of the difference between temperature in upper ocean layer and the freezing temperature in the upper 5 m-layer. The supercooled water is adjusted based on the freezing potential to form new ice and rejects brine into the ocean that leads to saltier water between 20-50 m in Figure R8. These analyses and texts were added in the revised manuscript.**

[Figure]

**Figure R7 Time-series of sea ice mass budget terms for (a) Y21_VT and (b) Y21_RP.**

[Figure]

**Figure R8 (a) the average temperature profile of upper 150 m under ice-covered areas for the difference between Y21_RP and Y21_VT. (b) same as (a), but for the salinity profile.**

- Little differences between sea ice model experiments. It is worth to document the effect but it can be moved to supplementary.

**Response: Thanks for the reviewer's suggestion. In this revision, we added a supplementary material. We put the results contain very basic information (i.e., little differences) to the supplementary. In the main body, we focus on the analyses showing more concrete evidence of physical process linking improved/changed physical parameterizations to simulated sea ice state.**

Specific minor comments:

4. The terms of "Anomaly" and "bias" in Section 3 are used inappropriately with respect to the reference (observation/reanalysis) field in such a short time scale (i.e. monthly mean in 2018). "Prediction error" fits better in this context.

**Response: Thanks for the reviewer's comment. We changed related terms to "prediction error".**

5. Figure 7 d-i REF was not introduced.

**These are typos. We corrected them as shown in Figure R9.**

[Figure]

**Figure R9 ERA5 monthly mean of downward longwave radiation at the surface for (a) July, (b) August, and (c) September, the difference between Y20_MOD and ERA5 for (d) July, (e) August, (f) September, and the difference between Y21_CTRL and Y20_MOD for (g) July, (h) August, and (i) September.**

6. Line 325 (Figure 8) "Y21_RP also shows much better predictive skill after late August ...". I disagree with it when I look at the regional plots. The differences between Y21_RP and other two experiments are very small in three regions with the exception of larger positive errors in Y21_RP in BEA-CHU than the other two. The better match with observation in the total Arctic sea ice extent results from the dominant negative errors in all experiments compensated by the positive errors in the BEA-CHU.

For the same reason, " the good fit" of recovery of the total sea ice extent in autumn is mis-interpretated in Fig 13 in contrast to mismatch in each subregion.

**Response: We modified this part. Now it reads as "Y21_RP also shows better predictive skill after late August compared with the CLIM/DAMP predictions (black dashed and dotted lines). This suggests the delayed ice recovery in late September shown in Y20_MOD, Y21_CTRL and Y21_VT is in part due to the choice of ocean advection and vertical mixing schemes, which change the behavior of ocean state. At the regional scale, the slower ice decline after July and earlier recovery of the ice extent in September mainly occur in the Beaufort-Chukchi and Barents-Kara-Greenland Seas compared to that of Y21_CTRL (Fig. 4a, c)." We agree with the reviewer that the better fit of the total sea ice extent might be a result of the compensating error in the subregions. Thus we added the following texts "The results of prediction experiments show that the updated Coupled Arctic Prediction System with improved physical parameterizations can better predict the evolution of the total sea ice extent compared with its predecessor described in Yang et al. (2020), though the predictions exhibit some biases in regional ice extent."**

7. Line 400 "significant influences" is weak without sufficient sampling.

**Response: Thanks for the reviewer's comment. In the revision, we moved this part to the supplementary and just focused on physical process linking improved/changed physical parameterizations to large changes in simulated sea ice state.**

Reference:

Armitage, T. W., Manucharyan, G. E., Petty, A. A., Kwok, R., and Thompson, A. F.: Enhanced eddy activity in the Beaufort Gyre in response to sea ice loss, Nat. Commun., 11, 1–8, 2020.

Dai, P., Gao, Y., Counillon, F., Wang, Y., Kimmritz, M., and Langehaug, H. R.: Seasonal to

decadal predictions of regional Arctic sea ice by assimilating sea surface temperature in the Norwegian Climate Prediction Model, Clim. Dynam., 54, 3863–3878,https://doi.org/10.1007/s00382-020-05196-4, 2020.

**Response to comments by Reviewer #2**

**We would like to thank the reviewer for the helpful comments on the paper.**

General comments

This study evaluates modeled summertime sea ice evolution in a updated Arctic regional coupled modeling system based on WRF, ROMS and CICE. A series of experiments with varied physical options has been conducted. These physical options include utilizing RAP physics in WRF, changing vertical coordinate transformation and stretching function and advection scheme in ROMS, involving MUSHY thermodynamics in CICE. Additional experiments have been focused on impacts of data assimilation with different radius of influence and ice thickness merging algorithm. The authors also discussed the model performance on timescale up to 7 months. However, the main limitation of this study is lack of novelty. Since the upgrades of physical parameterization in component models are achievements of community efforts, this study fails to introduce unique scientific contribution to the modeling community. The reported results in this study are general, detailed analysis of physical process linking selected physical parameterization to modeled sea ice state are missing. The discussion and conclusion sections just present the model results without adequate discussions in deep, mainly suffering from a lack of detailed physical process analysis in the previous sections. Based on my evaluation, I recommends resubmission after substantial improvement and scientific significance have added into this study.

Specific comments

Line 48-59: The authors stated that the gap of predictive skill between GCMs and "perfect model" may be related to inaccurate initial conditions and/or inadequate physical parameterizations. Please clarify in detail: what defines GCM and what defines "perfect model".

**Response: "GCMs" is referred to "global climate models". To avoid the confusion, we changed "GCMs" to "global climate models" in the revised manuscript. A "perfect**

model approach" treats one member of an ensemble as the truth (i.e., assuming the model is prefect without bias) and analyzes the skill of other members in predicting the response of the "truth" member (e.g., Meehl et al., 2007). We also changed the relevant texts in the revised manuscript.

*Meehl, G.A., Stocker, T. F., Collins, W. D., et al.: Global Climate Projections. In: Climate Change 2007: The Physical Science Basis. Contribution of Working Group I to the Fourth Assessment Report of the Intergovernmental Panel on Climate Change [Solomon, S., D. Qin, M. Manning, Z. Chen, M. Marquis, K.B. Averyt, M. Tignor and H.L. Miller (eds.)]. Cambridge University Press, Cambridge, United Kingdom and New York, NY, USA, 2007.*

Line 117-121: The authors stated that the change of coupling strategy in the latest ROMS model prevents the potentially erroneous results when the ROMS timestep is smaller than the coupling frequency with other model components. This is hard to follow. Please explain.

**Response: In the old version of the ROMS, the downward shortwave/longwave radiation from the WRF model is saved into two arrays ($srflx$ and $lrflx$). However, the bulk-flux algorithm in the old ROMS updated $srflx$ and $lrflx$ by subtracting the upward shortwave/longwave radiation without using independent arrays to save the net shortwave/longwave radiation. When the ROMS has shorter time-step than the coupling frequency (i.e., 5min vs. 30min), the upward shortwave/longwave radiation were subtracted from $srflx$ and $lrflx$ multiple times without updating the information from the WRF. In fact, we have already fixed this issue in Y20 before this study. To avoid the confusion, we removed the relevant text in the revised manuscript, and focused on technical details about the improved physical parameterizations.**

Figure 3: Since the configurations of Y21_CTRL and Y20_MOD are identical except model physics. The evolution of red and blue lines should start from the same point and then diverge. Please modify the relative figures.

**Response: Thanks for the reviewer's comment. This is because that the initial ensembles are generated by applying the second-order exact sampling (Pham, 2001) to sea ice state vectors (ice concentration and thickness) from an one-month free run of the coupled modeling system. The difference in the initial ice extent is due to that sea ice fields in Y20_MOD and Y21_CTRL (as well as other experiments listed in Table 2) are initialized based on one-month free runs (section 2), which use different physical configurations listed in Table 2. These one-month free runs do not have the same**

**evolution in sea ice state vectors, which result in different initial sea ice fields after data assimilation. We added this in the revision.**

Line 217-221: "Compared with the CLIM/DAMP predictions, both Y20_MOD and Y21_CTRL have smaller biases after early August." This statement is true in part. The authors seem ignore the sea ice extent evolutions in early September, since their biases are comparable. Again, "At the regional scale, in the Beaufort-Chukchi Seas, Y21_CTRL predicts slower ice retreat after late July than that of Y20_MOD, whereas in the East Siberian-Laptev Seas, Y20_MOD shows slower ice decline after mid-July than that of Y21_CTRL." This statement is true, but a further comparison with NSIDC evolution is missing. From Figure 3a and 3b, the performance of Y20_MOD is better than Y21_CTRL.

**Response: Thanks for the reviewer's comments. We revised the related texts to "Compared with the CLIM/DAMP predictions (black dashed and dotted lines), both Y20_MOD and Y21_CTRL have smaller biases in August, but comparable biases after early September.". "The difference in sea ice extent becomes larger at regional scales, in the East Siberian-Laptev Seas, Y21_CTRL shows faster ice decline after mid-July than that of Y20_MOD, whereas in the Beaufort-Chukchi Seas, Y21_CTRL predicts slower ice retreat after late July than that of Y20_MOD (Fig. 4a, 4b). Both Y20_MOD and Y21_CTRL agree well with the observations in the Barents-Kara-Greenland Seas (Fig. 4c). Compared with the observations, Y20_MOD performs relatively better in regional ice extents than that of Y21_CTRL.".**

Line 222-226: The authors attribute the underestimation of sea ice extent in BAY-CAA in both experiments to the difference in land/sea mask between the model and NSIDC grid. I am not convinced by this sentence. I don't find any information about the NSIDC grid or land/sea mask difference between model and NSIDC grid in the context.

**Response: Thanks for the reviewer's comment. We re-examined the attribution of the underestimated sea ice extent in the Canadian Archipelago and Baffin Bay region. It is largely due to our Coupled Arctic Prediction System simulates less sea ice in this region as shown in Figure R9. The inconsistent land/sea mask plays a secondary role. We modified the text to reflect this.**

[Figure]

**Figure R9 Monthly mean of sea ice concentration for (a, d, g) July, (b, e, h) August, (c, f, i) September of the NSIDC observations and Y21_CTRL, and the difference between Y21_CTRL and the NSIDC observations.**

Line 231-234: This result is not intuitive from Figure 4d-i. Additional subplots showing deviation between Y21_CTRL and Y20_MOD is needed.

**Response: Thanks for the reviewer's suggestion. In this revision, we added the figure showing the difference between Y21_CTRL and Y20_MOD (Fig. R5). It shows that Y21_CTRL predicts lower (higher) ice concentration along the East Siberian-Laptev (Beaufort-Chukchi) Seas (Fig. 5e₁-e₃). Y21_CTRL also predicts less ice in the central Arctic Ocean in August and September.**

[Figure]

**Figure R5 Monthly mean of sea ice concentration for (a) July, (b) August, (c) September of the NSIDC observations, and the difference between the all prediction experiments and the observations for (d₁-g₁) July, (d₂-g₂) August, (d₃-g₃) September. Vertical/horizontal-line areas represent the difference of ice edge location (15% concentration).**

Line 266-267: "It also shows that the magnitude of biases decreases as the lead time decreases". This sentence is not clear. Please revise.

**Response: Thanks for the reviewer's comment. We revised the sentence to "It appears that the magnitude of the biases tends to decrease over the areas with large biases as the prediction time increases (i.e., July vs. September)."**

Line 298-305: In the two vertical coordinate transformations, hc utilizes two values: 10 m and 300 m. Please present the reason why and how these two values are decided.

**Response: The choice of hc in Y20_CTRL (10 m) is the inherent limitation of the vertical transformation 1, in which hc must be less than or equal to the minimum value**

of water depth. As a result, hc was chosen as 10 m due to the limitation of the minimum value of water depth in Y20. With the vertical transformation 2, hc can be any positive value and expected to be the thermocline depth. Also, the choice of hc controls the vertical coordinate as "z-like" or "sigma-like" coordinate above the hc (e.g., Shchepetkin and McWilliams, 2005, Fig. 1b-c). As a result, we chose 300 m for Y21_VT. Compared to Y21_CTRL, Y21_VT is less sensitive to the bathymetry and its layers are more evenly-distributed in the upper 300 meters (Figure R11 vs. Figure R10). We modified the text to make it clear.

[Figure]

**Figure R10** The vertical layer distribution of Y21_CTRL for a cross section near the central Arctic.

[Figure]

**Figure R11** The vertical layer distribution of Y21_VT for a cross section near the central Arctic.

*Shchepetkin, A. F., and McWilliams, J. C.: The Regional Ocean Modeling System: A split-explicit, free-surface, topography following coordinates ocean model, Ocean Modelling, 9, 347-404, 2005.*

Figure 10: The figure is too small. It is better to arrange the two subplots into one column.

**Response: Thanks for the reviewer's comment. We modified the figure as shown in Figure R12.**

[Figure]

**Figure R12 First column: monthly mean of sea surface temperature for (a) July, (b) August, (c) September of the OI SST. Second column: the difference between Y21_CTRL and the OI SST for (d) July, (e) August, (f) September. Right panel: Monthly mean of sea surface temperature difference between Y21_VT/Y21_RP and Y21_CTRL for (g) July, (h) August, (i) September of Y21_VT, (j) July, (k) August, and (l) September of Y21_RP.**

Figure 12: The second row is not needed.

**Thanks for the reviewer's suggestion. We modified the figure and only keep the difference as shown in Figure R13.**

[Figure]

**Figure R13 Monthly mean of sea ice thickness difference between Y21_MUSHY and Y21_RP for (a) July, (b) August, and (c) September.**

---

## Referee Report (RR1)

General comments

This is my second time to review this manuscript. Comparing to the previous version, the authors have done a lot of works to address my major concerns and those of the other reviewer. Specifically, many details relating to the involved parameterizations in WRF, ROMS and CICE, the changes in ice, atmospheric and oceanic states, and the ice mass budget analysis in CICE have been added into the revised manuscript. Based on my evaluation, now the manuscript could be done as a minor revision.

Specific comments

Line 204-208: The description of "ice mass budget analysis" in CICE is quite brief. The authors should introduce it in details because of that lots of the following paragraphs relate to the method.

Line 262: the authors should notice that Y21_CTRL generates colder bias (worse result) than Y20_MOD in the central Northern Atlantic in August, as a complementary to the better result of "especially over eastern Siberia and the Atlantic sector in July to September "

Line 297: change Y20_CTRL to Y21_CTRL and Y21_MOD to Y20_MOD

Line 321-323: This is a comment (NO need to reply). Atmospheric heat flux at ice surface and oceanic heat flux at ice bottom contribute more "directly" to sea ice area and sea ice thickness change, rather than sea ice extent change, as sea ice extent change is also affected by wind forcing and ocean currents.

Line 393: why there is frazil ice formation in July? This is quite anti-intuitive. Line 396-408 give some explanations on it, but I would like to say that it is a "purely" model adjustment, I am not sure whether this will happen in the real ocean.

Line 468-486: the description of Y21_MAR-7 experiment does not present useful information or novelty to the manuscript. I suggest the authors to consider removing the part.

---

## Author Response (AR2)

**Response to the reviews of** "An improved regional coupled modeling system for Arctic sea ice simulation and prediction: a case study for 2018" by Chao-Yuan Yang, Jiping Liu, Dake Chen

**Response to comments by Reviewer #2**

**We would like to thank the reviewer for the helpful comments on the paper.**

General comments:

This is my second time to review this manuscript. Comparing to the previous version, the authors have done a lot of works to address my major concerns and those of the other reviewer. Specifically, many details relating to the involved parameterizations in WRF, ROMS and CICE, the changes in ice, atmospheric and oceanic states, and the ice mass budget analysis in CICE have been added into the revised manuscript. Based on my evaluation, now the manuscript could be done as a minor revision.

Specific comments:

Line 204-208: The description of "ice mass budget analysis" in CICE is quite brief. The authors should introduce it in details because of that lots of the following paragraphs relate to the method.

**Response: Thanks for the reviewer's suggestion. In this revision, we added more descriptions of the sea ice mass budget analysis. Now it reads "In order to understand physical contributors that drive the evolution of Arctic sea ice state (the standard variables of the ice concentration and thickness), the mass budget of Arctic sea ice for all experiments is analyzed in this study as defined in Notz et al. (2016, Append. E), including:**

- **sea ice growth in supercooled open water (frazil)**

- **sea ice growth at the bottom of the ice (basal growth)**

- **sea ice growth due to transformation of snow to sea ice (snowice)**

- **sea ice melt at the air-ice interface (top melt)**

- **sea ice melt at the bottom of the ice (basal melt)**

- **sea ice melt at the sides of the ice (lateral melt)**

- **sea ice mass change due to dynamics-related processes (e.g. advection) (dynamics)**

**These diagnostic variables are determined by saving the ice mass tendency of above processes separately every time step and integrated to output the daily-mean value."**

Line 262: the authors should notice that Y21_CTRL generates colder bias (worse result) than Y20_MOD in the central Northern Atlantic in August, as a complementary to the better result of "especially over eastern Siberia and the Atlantic sector in July to September"

**Response: Thanks for the reviewer's comment. In this revision, we modified the text to reflect this. "The Y21_CTRL experiment with the RAP physics tends to reduce the prediction errors in Y20_MOD, especially over eastern Siberia and the Atlantic sector in July to September (Fig. 2g-i). However, Y21_CTRL results in larger bias in the central Northern Atlantic in August than that of Y20_MOD (Fig. 2h)".**

Line 297: change Y20_CTRL to Y21_CTRL and Y21_MOD to Y20_MOD

**Response: Thanks. We corrected them.**

Line 321-323: This is a comment (NO need to reply). Atmospheric heat flux at ice surface and oceanic heat flux at ice bottom contribute more "directly" to sea ice area and sea ice thickness change, rather than sea ice extent change, as sea ice extent change is also affected by wind forcing and ocean currents.

**Response: We agree with the reviewer.**

Line 393: why there is frazil ice formation in July? This is quite anti-intuitive. Line 396-408 give some explanations on it, but I would like to say that it is a "purely" model adjustment, I am not sure whether this will happen in the real ocean.

**Response: We agree with the reviewer that frazil ice formation in July is more likely to be the results of model adjustment and/or numerical error. We added this in the revised manuscript. Besides some explanations on L396-408, we also provided extended discussions for the spurious frazil ice formation in section 5. In L539-L558, we mentioned**

that the treatment of ice-ocean heat flux parameterization may play a role in the spurious frazil ice formation if the ice-ocean heat flux reaches the limit imposed by the melting potential. This implies the ocean surface layer will be close to the freezing temperature. The ice-ocean heat flux combined with the oscillatory behaviors of advection schemes will further increase the amount of frazil ice.

Line 468-486: the description of Y21_MAR-7 experiment does not present useful information or novelty to the manuscript. I suggest the authors to consider removing the part.

**Response: Based on the reviewer's suggestion, we removed the description related to the Y21_MAR-7 experiment in the revised manuscript.**

**Response to comments by Topical editor**

**We would like to thank the editor for the helpful comments on the paper.**

Comments to the author:

The reviewer is happy with your revision and has a few minor comments. In addition, I have two minor comments:

(1) Please revise the figure captions so that readers can largely understand the figures without reading the paper text and without searching/guessing the meaning of the experiment names in the captions.

**Response: Thanks for the editor's helpful suggestions. We modified the figure captions by briefly describing changes made in each experiment. For example:**
**Figure 4 Top panel: Time-series of Arctic sea ice extent for the observations (black line) and the ensemble-mean of Y20_MOD (blue line, the original CAPS), Y21_CTRL (yellow line, changes in the atmospheric physics), Y21_VT (red line, changes in the ocean vertical coordinate), Y21_RP (green line, changes in the oceanic advection), and Y21_MUSHY (pink line, changes in sea ice thermodynamics). Dashed and dotted lines are the climatology and the damped anomaly persistence prediction. Bottom panel: Time-series of the observed (black line) and the ensemble-mean of regional sea ice extents for Y20_MOD (blue line), Y21_CTRL (yellow line), Y21_VT (red line), Y21_RP (green line), and Y21_MUSHY (pink line) for (a) Beaufort-Chukchi Seas, (b) East Siberian-Laptev**

**Seas, and (c) Barents-Kara-Greenland Seas.**

(2) Please do a thorough English check. Here I just provide a few examples of English issues in the "abstract":

L16 by--> using

L17-20, , which ... --> . CAPS is built on ...

L20 process --> processes

L23 improved simulation in --> an improved simulation of

L25 reduces --> reduce, changes --> change

L30 have --> has

L32 making --> to make

**Response: Thanks for the editor's comments. We changed them and did a thorough English check in the revised manuscript.**